# You Only Debias Once: Towards Flexible Accuracy-Fairness Trade-offs at Inference Time

## Abstract

Deep neural networks are prone to various bias issues, jeopardizing their applications for high-stake decision-making. Existing fairness methods typically offer a fixed accuracy-fairness trade-off at inference time , since the weight of the well-trained model is a fixed point (fairness-optimum) in the weight space [1]. Nevertheless, more flexible accuracy-fairness trade-offs at inference time are practically desired since: 1) stakes of the same downstream task can vary for different individuals, and 2) different regions have diverse laws or regularization for fairness. If using the previous fairness methods, we have to train multiple models, each offering a specific level of accuracy-fairness trade-off. This is often computationally expensive, time-consuming, and difficult to deploy, making it less practical for real-world applications. To address this problem, we propose *You Only Debias Once* (YODO) to achieve in-situ flexible accuracy-fairness trade-offs at inference time, using *a single model* that trained only once [2]. Instead of pursuing one individual fixed point (fairness-optimum) in the weight space, we aim to find a "line" in the weight space that connects the accuracy-optimum and fairness-optimum points using a single model. Points (models) on this line implement varying levels of accuracy-fairness trade-offs. at inference time, by manually selecting the specific position of the learned "line", our proposed method can achieve arbitrary accuracy-fairness trade-offs for different end-users and scenarios. Experimental results on tabular and image datasets show that YODO achieves flexible trade-offs between model accuracy and fairness, at ultra-low overheads. Our codes are anonymously available at https://anonymous.4open.science/r/yodo-BB81.

To Reviewer bMqY: Requested Change 1: We revise "the inference time" to "inference time" throughout the paper. And we only highlight this one.

## 1 Introduction

Deep neural networks (DNNs) are prone to be biased with respect to sensitive attributes (Mehrabi et al., 2021; Du et al., 2020; Dwork et al., 2012; Binns, 2018; Caton & Haas, 2020; Barocas et al., 2017), which raises concerns about the application of deep learning model on high-stake decision-making, such as credit scoring (Petrasic et al., 2017), criminal justice (Berk et al., 2021), job market (Hu & Chen, 2018), healthcare (Rajkomar et al., 2018; Ahmad et al., 2020; Bjarnadóttir & Anderson, 2020; Grote & Keeling, 2022) and education Bøyum (2014); Brunori et al. (2012); Kizilcec & Lee (2022). Decisions made by these biased algorithms could have a long-lasting even life-long impact on people's lives and may affect underprivileged groups negatively. This concern about biased models has aroused wide interest from both academic and industrial researchers in analyzing and achieving fairness for DNNs.

Many studies have shown that achieving fairness in deep learning models involves a trade-off with model accuracy (Bertsimas et al., 2011; Menon & Williamson, 2018; Zhao & Gordon, 2019; Bakker et al., 2019). This has led to the development of various methods aimed at addressing fairness in deep learning models (Edwards & Storkey, 2015; Louppe et al., 2017; Chuang & Mroueh, 2020), but typically these methods have been designed to achieve a fixed level of accuracy-fairness trade-off. In real-world applications, however, the

---

[1]The *weight space* of a model refers to the space of all possible values that the model's trainable parameters can take. If a neural network has $m$ trainable parameters, then the dimension of the weight space is $m$. Each point, a $m$-dimensional vector, in the weight space corresponds to a specific model. For visualization purposes, the weight space is often reduced to a 2D space, as shown in Figure 1.

[2]This is also what we mean by *one-time training* in this paper.

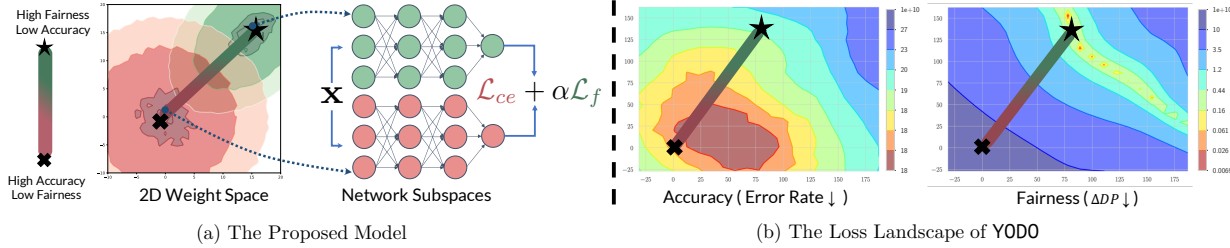

(a) The Proposed Model    (b) The Loss Landscape of YODO

Figure 1: **(a)**:The overview of our proposed method. 2D Weight Space indicated the landscape of model accuracy and fairness. ✖ indicates the accuracy-optimum weight with high accuracy but low fairness, and ★ indicates the fairness-optimum weight with low accuracy but high fairness. Network Subspaces shows the different subspaces correspond with different objectives (i.e., accuracy $\mathcal{L}_{ce}$ and fairness $\mathcal{L}_f$). **(b)**: The loss landscape of the model accuracy (error rate) and fairness ($\Delta DP$) in the same weight space of our proposed method. The weight space is reduced to two dimensions (Garipov et al., 2018). The different points indicate different objectives, ✖ indicates the accuracy–optimum endpoint in the weight space, while ★ indicates the fairness-optimum endpoint in the weight space. The dataset is ACS-I with gender as sensitive attribute.

appropriate trade-off between accuracy and fairness may vary depending on the context and the needs of different stakeholders/regions. Thus, it is important to have flexible trade-offs at inference time due to:

**1) Downstream tasks with different stakes can have varying fairness requirements, depending on the individuals involved.** According to a survey by Srivastava et al. (2019), people prioritize accuracy over fairness when the stakes are high in certain domains. For example, in healthcare (e.g., cancer prediction), accuracy should be favored over fairness, as prioritizing fairness can lead to "fatal" consequences, such as missing cancer diagnoses at higher rates (Chen et al., 2018; Srivastava et al., 2019). Since the stakes of downstream tasks differ among individuals, the expected level of fairness may also vary significantly. Thus, in high-stakes domains, the trade-off between accuracy and fairness should be flexible and controllable at inference time.

**2) Different regions have diverse laws or regulations for fairness.** The use of decision-making systems is regulated by local laws and policies. However, countries may exhibit differences in the importance of fairness in various applications. For example, the labor market and employees may expect fairness to be much stronger in Germany than in North America (Gerlach et al., 2008). Therefore, developers of machine learning models should consider the varying fairness requirements when applying their algorithms in different regions.

Unfortunately, while urgently needed, flexible trade-offs between model accuracy and fairness at inference time remain underexplored. Thus, we aim to answer the following question in this paper:

*Can we achieve flexible accuracy-fairness trade-offs at inference time using a single model that trained only once?*

It is an open but challenging problem to achieve flexible accuracy-fairness trade-offs. One solution is to train multiple models for different trade-offs, which limits its practical use due to the significant training time and memory overhead. Another alternative solution is the post-processing method for fairness, which may lead to suboptimal model accuracy and require sensitive attributes at inference time (Woodworth et al., 2017).

To address the above question, we propose *You Only Debias Once* (YODO) to achieve flexible fairness-accuracy trade-offs via learning an *objective-diverse* neural network subspace that contains accuracy-optimum and fairness-optimum points in the weight space. As illustrated in Figure 1(a), we design an objective-diverse neural network subspace, which contains two endpoints (the red network and the green network in the weight space.) The two endpoints[3] are encouraged to accuracy-optimum (✖ in Figure 1) and fairness-optimum (★ in Figure 1) solutions, respectively. During the training time, we encourage the two endpoints to converge to accuracy-optimum and fairness-optimum points in the weight space, and also encourage the "line" between the two endpoints to achieve transitional solutions. Specifically, we combine the two endpoints with random weight and then optimize them with a corresponding combination of accuracy and fairness objectives. at inference time, we enable the model to achieve arbitrary accuracy-fairness trade-offs with the trained model

---

[3]One endpoint represents a specific model parameter.

by manually selecting a trade-off coefficient to determine the model weight (namely, select a position of the line). Our **contributions** are highlighted as follows:

- We propose a new target, achieving in-situ flexible accuracy-fairness trade-offs at inference time while only trained once, which adaptively meets the requirements of flexible fairness in real-world applications.

- We achieve the above in-situ flexible accuracy-fairness trade-offs by introducing an **objective-diverse** neural network subspace to achieve accuracy-fairness trade-offs. The subspace has two different endpoints in weight space, which are optimized for accuracy-optimum and fairness-optimum at different endpoints of subspaces and the "line" between the two endpoints to achieve transitional solutions. Thus it can achieve flexible trade-offs by customizing the endpoints at inference time, with ultra-low overheads.

- Experimental results validate the effectiveness and efficiency of the proposed method on both tabular and image data. The result of experiments shows that our proposal achieves comparable performance with only trained once with various neural network architectures when compared to trained models for single use or fairness on a single attribute only. The visualization experiment provides the insight that our method gradually learns fair representation, which guarantees fair prediction.

## 2    Related Works

To Reviewer bMqY: Concern 5: we move literature review earlier.

In this section, we discuss three areas of research that are related to our work, namely group fairness, accuracy-fairness trade-offs, and neural network subspace learning.

**Fairness in Machine Learning.** Recently, algorithmic fairness (Pogodin et al., 2023; Cruz et al., 2023; Guo et al., 2023; Li et al., 2023; Pham et al., 2023; Jung et al., 2023; Han et al., 2023; Chung et al., 2023; Roh et al., 2021; Guldogan et al., 2023; Roh et al., 2023; Schrouff et al., 2022a;b; Vogel et al., 2020; Coston et al., 2020; Maheshwari et al., 2022) is required legally or morally in machine learning systems and various fairness definitions in machine learning systems have been proposed to meet the requirement of fairness expectation. The fairness can typically be classified into *individual fairness* or *group fairness*, which can be achieved via pre/in/post-processing. In this paper, we focus on the in-processing group fairness, which measures the statistical parity between subgroups defined by the sensitive attributes, such as gender or race (Zemel et al., 2013; Louizos et al., 2015; Hardt et al., 2016; Chuang & Mroueh, 2020; Zafar et al., 2017; Madras et al., 2018; Joseph et al., 2016). Nevertheless, these constraints are trained and satisfied during training, the model may expect different accuracy-fairness trade-offs at inference time. In contrast, Our proposed method, YODO, aims to address this issue by enabling flexible accuracy-fairness trade-offs at inference time.

**Accuracy-Fairness Trade-offs.** Many existing works investigate the trade-offs between the model accuracy and fairness(Cooper et al., 2021; Kim et al., 2020; Bertsimas et al., 2011; Menon & Williamson, 2018; Liu & Vicente, 2020; Kleinberg, 2018; Haas, 2019; Rothblum & Yona, 2021; Wei & Niethammer, 2020; Barlas et al., 2021; Dutta et al., 2020; Maity et al., 2020; Chouldechova & Roth, 2018; Zhang et al., 2022; Vogel et al., 2021). Maity et al. (2020) discusses the existence of the accuracy-fairness trade-offs. (Kim et al., 2020; Dressel & Farid, 2018; Zliobaite, 2015; Dutta et al., 2020; Blum & Stangl, 2019; Wick et al., 2019) shows that fairness and model accuracy conflict with each another, and achieved fairness often comes with a necessary cost in loss of model accuracy. Cooper et al. (2021); Kim et al. (2020) re-examines the trade-offs and concludes that unexamined assumptions may result in emergent unfairness. These studies emphasize the need to develop models that balance accuracy and fairness while also considering the unique characteristics of the data and the potential for emergent biases. Navon et al. (2021) proposed Pareto-Front Learning (PFL), which employs Pareto HyperNetworks (PHNs) for efficient and scalable learning of entire Pareto fronts in multi-objective optimization, enabling post-training selection of desired operating points. Mehta et al. (2022) proposed a reinforcement learning approach (PA-Net) that efficiently approximates Pareto fronts in bi-objective travelling salesperson problems, enhancing performance and application in robotic navigation tasks. Menon & Williamson (2018) examines fairness tradeoffs in binary classifiers, linking fairness measures to cost-sensitive risks and revealing that optimal classifiers use instance-dependent thresholding, with a proposed simple approach for fairness-aware problems. Kim et al. (2020); Liu & Vicente (2022); Wick et al. (2019); Dutta et al. (2020) discuss the trade-offs between accuracy and fairness.

To Reviewer 21XQ: Requested Change 4: discussion on accuracy-fairness trade-off.

**Neural Network Subspaces Learning.** The idea of learning a neural network subspace is concurrently proposed by Wortsman et al. (2021) and Benton et al. (2021). Multiple sets of network weights are treated as the corners of a simplex, and an optimization procedure updates these corners to find a region in weight space in which points inside the simplex correspond to accurate networks. Garipov et al. (2018) learning a connection between two independently trained neural networks, considering curves and piecewise linear functions with fixed endpoints. Wortsman et al. (2021) and Benton et al. (2021) concurrently proposed to learn a functionally diverse subspace. Our proposed method YODO differs from these methods by specifying each subspace to the different learning objectives and allowing flexible fairness levels at inference time. Dosovitskiy & Djolonga (2020) trains a single model on distribution of losses, allowing it to generate outputs corresponding to any loss from the training distribution, increasing efficiency at both training and inference times, as demonstrated in beta-VAE, learned image compression, and fast style transfer tasks. Ha et al. (2017) proposed Hypernetworks, which generate weights for another network, achieve competitive results in sequence modeling and image recognition tasks with fewer learnable parameters, challenging traditional weight-sharing paradigms.

> **To Reviewer 21XQ: Requested Change 4: add more discussion on accuracy-fairness trade-off.**

## 3 Preliminaries

In this section, we introduce the notation used throughout this paper and provide an overview of the preliminaries of our work, including algorithmic fairness and neural network subspace learning.

**Notations.** We use $\{\mathbf{x}, y, s\}$ to denote a data instance, where $\mathbf{x} \in \mathbb{R}^d$, $y \in \{0, 1\}$, $s \in \{0, 1\}$ are feature, label, sensitive attribute, respectively. $\hat{y} \in [0, 1]$ denotes the predicted value by machine learning model $\hat{y} = f(\mathbf{x}; \theta) : \mathbb{R}^d \to [0, 1]$ with trainable parameter $\theta \in \mathbb{R}^m$ in the $m$-dimensional weight space, which is represented as a $m$-dimensional flatten vector. $\mathcal{D}$ denotes the data distribution of $(\mathbf{x}, y)$ and $\mathcal{D}_0 / \mathcal{D}_1$ denotes the distribution of the data with sensitive attribute $0/1$. In this work, we consider the fair binary classification ($y \in \{0, 1\}$) with binary sensitive attributes ($s \in \{0, 1\}$).

### 3.1 Algorithmic Fairness

For simplicity of the presentation, we focus on the group fairness definition of demographic parity (DP) (Dwork et al., 2012). DP is a fairness metric that aims to ensure that the proportion of positive outcomes is equal between different demographic groups in the population. To measures the DP between two demographic groups, $\Delta DP$ is defined as the absolute difference in the positive prediction rates of the machine learning model between the two demographic groups. This definition is proposed to ensure that the model treats different groups equally. Demographic parity (DP) is defined as follows:

> **To Reviewer 21XQ: Requested Change 1: We remove "without loss of generality".**

**Definition 3.1** (Demographic Parity). *Demographic Parity requires the predicted values $\hat{y}$ to be independent of the sensitive attribute $s$, that is, $P(\hat{y}|s = 0) = P(\hat{y}|s = 1)$.*

One relaxed metric to measure demographic parity has also been proposed by Edwards & Storkey (2015) and widely used by Agarwal et al. (2018); Wei et al. (2019); Taskesen et al. (2020); Madras et al. (2018); Chuang & Mroueh (2020). The relaxed metric $\Delta DP$ of demographic parity is defined as follows:

$$\Delta\mathrm{DP}(f) = \left| \mathbb{E}_{\mathbf{x} \sim \mathcal{D}_0} f(\mathbf{x}) - \mathbb{E}_{\mathbf{x} \sim \mathcal{D}_1} f(\mathbf{x}) \right|, \tag{1}$$

One practical and effective way to achieve demographic parity is to formulate a penalized optimization with $\Delta DP$ as gap regularization. The overall objective for a fixed level of accuracy-fairness trade-offs is as follows:

$$\mathcal{L} = \mathcal{L}_{ce}(f(\mathbf{x}; \theta), y) + A \cdot \mathcal{L}_f(f(\mathbf{x}; \theta), y) = \mathcal{L}_{ce} + A \cdot \mathcal{L}_f, \tag{2}$$

where $\mathcal{L}_{ce}$ is the objective function (e.g., cross-entropy) of the downstream task, $\mathcal{L}_f$. is instantiated as the demographic parity difference $\Delta DP(f)$, and $A$ is a fixed hyperparameter to balance the model accuracy and fairness. In the following, we set $A$ to 1 for simplicity. We also conducted experiments to explore the effect of varying $A$ in Appendix C.3.

> **To Reviewer zQ7a: Concern 3: explicitly introduce $\mathcal{L}_f$.**

In addition to DP, we also consider the fairness metric of Equality of Opportunity (EO) and Equalized Odds (Eodd) in our experiments, as described in Section 6.6 and Appendix C.2. Through the experiment on multiple fairness definitions (i.e., DP, EO, Eodd), we can gain a more comprehensive understanding of the performance of our approach and the trade-offs between fairness and accuracy.

> **To Reviewer DMiX: Adding more fairness criteria.**

## 3.2 Neural Network Subspaces

The optimization of the neural network is to find a minimum (often a local minimum) in a high-dimensional weight space. Wortsman et al. (2021) and Benton et al. (2021) proposed a method to learn a functionally diverse neural network subspace, which is parameterized by two sets of network weights, $\omega_1$ and $\omega_2$. At training time, a network is sampled from the line defined by this pair of weights, $\theta = (1 - \alpha)\omega_1 + \alpha\omega_2$ for $\alpha \in [0, 1]$, and the two sets of weights $\omega_1$ and $\omega_2$ are optimized by one back-propagation simultaneously, making the learning process more efficient. By parameterizing the neural network subspace using two sets of weights, this method allows for greater control over the optimization of the neural network. This is because the user can specify the values of $\alpha$ to control the degree of mixing between $\omega_1$ and $\omega_2$. Different from this method that the learning objective of both $\omega_1$ and $\omega_2$ are model accuracy for the downstream task, our proposed method lets the $\omega_1$ and $\omega_2$ learn different target, accuracy and fairness, respectively. The method also allows for the exploration of the trade-offs between accuracy and fairness, as the two sets of weights can be optimized to achieve different levels of accuracy and fairness.

## 4 Methodology

In this section, we introduce our method YODO, which learns an objective-diverse neural network subspace targeting flexible trade-offs between accuracy and fairness at inference time while training the model once.

### 4.1 You Only Debias Once

Our goal is to find an objective-diverse subspace (the "line") in weight space comprised of accuracy-optimum (✖ in Figure 1) and fairness-optimum (★ in Figure 1) neural networks, as illustrated in Figure 1. Specifically, we first parameterize two sets of trainable weights $\omega_1 \in \mathbb{R}^n$ and $\omega_2 \in \mathbb{R}^n$ for one neural network, and then optimize weights $\omega_1, \omega_2$ to achieve that $f(\mathbf{x}; \omega_1)$ is high for model accuracy and $f(\mathbf{x}; \omega_2)$ is high for model fairness. Thus $f(\mathbf{x}; (1 - \alpha)\omega_1 + \alpha\omega_2))$ achieves $\alpha$-controlled accuracy-fairness trade-offs. In other words, we aim to learn a "line" between $\omega_1$ and $\omega_2$ in the weight space to achieve flexible accuracy-fairness trade-offs at inference time.

In the training process, we aim to optimize $\omega_1, \omega_2$ with objective functions targeting accuracy and fairness objectives, respectively. Thus two endpoints $\omega_1$ and $\omega_2$ correspond with objective function for downstream task $\mathcal{L}_{ce}$ and the objective function for fairness. $\mathcal{L}_{ce} + \mathcal{L}_f$ [4]. Thus the learned $\omega_1, \omega_2$ will be accuracy-optimum (✖ in Figure 1) and fairness-optimum (★ in Figure 1) points in the weight space. Based on the linear combination of two endpoints, we try to minimize the loss as

$$\mathcal{L} = (1 - \alpha) \underbrace{\mathcal{L}_{ce}}_{\omega_1(\text{✖})} + \alpha \underbrace{(\mathcal{L}_{ce} + \mathcal{L}_f)}_{\omega_2(\text{★})} = \mathcal{L}_{ce} + \alpha\mathcal{L}_f, \tag{3}$$

where $\theta = (1 - \alpha)\omega_1 + \alpha\omega_2$ for $\alpha \in [0, 1]$. $\mathcal{L}_{ce}$ is instantiated as the cross-entropy loss for downstream tasks (i.e., binary classification task), and $\mathcal{L}_f$ is instantiated as demographic parity difference $\Delta\text{DP}[f(x; \theta)]$.

Since we seek to optimize the different levels of fairness constraints, for each $\alpha \in [0, 1]$, we propose to minimize $\mathbb{E}_{(\mathbf{x}, y) \sim \text{D}}[\mathcal{L}_{ce}(f(\mathbf{x}; \theta), y) + \alpha\mathcal{L}_f(f(\mathbf{x}; \theta), y)]$, thus our training objective is to minimize

$$\mathbb{E}_{\alpha \sim \text{U}}\big[\mathbb{E}_{(\mathbf{x}, y) \sim \text{D}}[\mathcal{L}_{ce}(f(\mathbf{x}; \theta), y) + \alpha\mathcal{L}_f(f(\mathbf{x}; \theta), y)]\big],$$
$$s.t. \quad \theta = (1 - \alpha)\omega_1 + \alpha\omega_2, \tag{4}$$

where U denotes the uniform distribution between 0 and 1 and the trade-off hyperparameter $\alpha$ follows uniform distribution, i.e., $\alpha \sim \text{U}$.

**Why Does YODO Achieve Flexible Accuracy-fairness Trade-offs in Terms of Objective Function?** For each $\alpha$ in U, the objective $\mathbb{E}_{(\mathbf{x}, y) \sim \text{D}}[\mathcal{L}_{ce}(f(\mathbf{x}; \theta), y) + \alpha\mathcal{L}_f(f(\mathbf{x}; \theta), y)]$ will be optimized to the minima, leading to different accuracy-fairness trade-offs with different $\alpha$. In other words, our model can be regarded

---

[4]For simplicity, here we set the hyperparameters $A$ in Equation (2) to 1.

as training infinite models with different fairness constraints in the training phase. Since our method aims to find the "line" between the accuracy-optimum and fairness-optimum points in the weight space, we need to ensure that any point on this line corresponds to a specific level of fairness. Therefore, we randomly sample a $\alpha$ during each epoch. In other words, each $\alpha$ corresponds to one model with a specific level of fairness. Under a wide range of different $\alpha$ values, we train numerous models (infinite) throughout the training process. Such a mechanism guarantees that YODO could achieve flexible trade-offs with one-pass training at inference time. In the following, we analyze our model from the model optimization perspective.

> **To Reviewer 21XQ: Concern 3**

### 4.2 Optimization Resulting in Objective-Diverse Subspace

In this section, we discuss the model optimization for YODO. In each training batch, we randomly sample $\alpha \sim U_{[0,1]}$, and then we use $\theta = (1-\alpha)\omega_1 + \alpha\omega_2$ as the model parameters. We calculate the gradients for $\omega_1$ and $\omega_2$ with respect to objective function (Equation (4)) as follows:

$$\frac{\partial \mathcal{L}}{\partial \omega_i} = \frac{\partial \mathcal{L}}{\partial \omega_i} = \frac{\partial \mathcal{L}}{\partial \theta} \frac{\partial \theta}{\partial \omega_i}. \tag{5}$$

Consider that $\theta = (1-\alpha)\omega_1 + \alpha\omega_2$, the gradients for the endpoints $\omega_1$ and $\omega_2$ are

$$\frac{\partial(\mathcal{L}_{ce} + \alpha\mathcal{L}_f)}{\partial \omega_1} = (1-\alpha)\frac{\partial(\mathcal{L}_{ce} + \alpha\mathcal{L}_f)}{\partial \theta},$$
$$\frac{\partial(\mathcal{L}_{ce} + \alpha\mathcal{L}_f)}{\partial \omega_2} = \alpha\frac{\partial(\mathcal{L}_{ce} + \alpha\mathcal{L}_f)}{\partial \theta}, \tag{6}$$

From Equations (5) and (6), we can see that gradients for $\omega_1$ and $\omega_2$ are related to the $\mathcal{L}_{ce}$ and $\mathcal{L}_f$ with different values of the scale coefficient (i.e., $1-\alpha$ and $\alpha$). The optimization of the two endpoints $\omega_1$ and $\omega_2$ only depends on the gradient $\frac{\partial(\mathcal{L}_{ce} + \alpha\mathcal{L}_f)}{\partial \theta}$, which are the most time-consuming operations in the back-propagation during the model training. This indicates that our method does not require any extra cost in the back-propagation phase since we only compute $\frac{\partial(\mathcal{L}_{ce} + \alpha\mathcal{L}_f)}{\partial \theta}$ once.

> **To Reviewer bMqY: Requested Change 1: We revise the $\omega_1$ and $\omega_2$ to the correct order. We only highlight this one, and we also the following content accordingly.**

**Why Does YODO Achieve Flexible Accuracy-fairness Trade-offs in Terms of Gradient?** Equation (6) indicates the optimization of the weights $\omega_1$ and $\omega_2$, as well as the objective function, are both controlled by the hyperparameter $\alpha$. In the following analysis, we will examine the optimization of these weights with different values of $\alpha$.

- When $\alpha = 0$, the gradients of $\omega_1$ in Equation (6) is calculated by the loss $\frac{\partial(\mathcal{L}_{ce} + \alpha\mathcal{L}_f)}{\partial \omega_1} = (1-0)\frac{\partial(\mathcal{L}_{ce} + 0*\mathcal{L}_f)}{\partial \theta} = \frac{\partial(\mathcal{L}_{ce})}{\partial \theta}$ and the gradients of $\omega_2$ is 0, indicating weight $\omega_1$ will be only optimized by the accuracy objective.

- When $\alpha = 1$, the gradients in Equation (6) is calculated by the loss $\frac{\partial(\mathcal{L}_{ce} + \alpha\mathcal{L}_f)}{\partial \omega_2} = 1*\frac{\partial(\mathcal{L}_{ce} + 1*\mathcal{L}_f)}{\partial \theta} = \frac{\partial(\mathcal{L}_{ce} + \mathcal{L}_f)}{\partial \theta}$ and the gradients of $\omega_1$ is 0, indicating weight $\omega_2$ will be only optimized by the fairness objective.

- When $0 < \alpha < 1$, the weights $\omega_1$ and $\omega_2$ will be only optimized by the linear combination of accuracy objective and fairness objective.

From the analysis of gradient, we can conclude that *the optimization tends to encourage the two endpoints $\omega_1$ and $\omega_2$ to accuracy-optimum and fairness-optimum solutions in the weight space.* As illustrated in Figure 1(b), we plot the landscape of the error rate (accuracy) and the $\Delta DP$ (fairness). The landscapes are depicted with a trained YODO. ✖ and ★ indicate the two endpoints, which are encouraged to learn accuracy and fairness objectives, respectively. The landscape shows that our method learns an objective-diverse neural network subspace and optimizes the endpoints to accuracy-optimum and fairness-optimum solutions.

**Model Optimization.** To promote the diversity of the neural network subspaces being learned, we follow the approach proposed in (Wortsman et al., 2021) and aim to learn two distinct endpoints, i.e., the accuracy-optimum and the fairness-optimum endpoint in the weight space. Specifically, we constrain the cosine similarity of the pair of endpoints ($\omega_1$ and $\omega_2$) to be zero. To achieve this, we augment the original learning

objective (Equation (4)) with a cosine similarity regularization term $\mathcal{L}_{reg}$, which is defined as $\mathcal{L}_{reg} = \frac{\langle \omega_1, \omega_2 \rangle^2}{|\omega_1|_2^2 |\omega_2|_2^2}$, which encourages the two endpoints to be as dissimilar as possible, effectively diversifying the learned subspaces. Minimizing $\mathcal{L}_{reg}$ (i.e., the cosine similarity of $\omega_1$ and $\omega_2$) promotes diversity between the two endpoints. Given that the cosine similarity of two random high-dimensional vectors

---

**Algorithm 1** YODO

---

**Require:** Training set $\mathcal{S}$, balance hyperparameters $\beta$
1: Initialize each $\omega_i$ independently.
2: **for** batch in $\mathcal{S}$ **do**
3:      Randomly sample an $\alpha$
4:      Calculate interpolated weight $\omega =\leftarrow (1 - \alpha)\omega_1 + \alpha\omega_2$
5:      Calculate loss $\mathcal{L} = \mathcal{L}ce + \alpha\mathcal{L}f + \beta\mathcal{L}_{reg}$
6:      Back-propagate $\mathcal{L}$ to update $\omega_1$ and $\omega_2$ using Equation (6).
7: **end for**

---

is typically close to zero, and considering that the accuracy-optimum and fairness-optimum points in the weight space are generally distinct, chasing $\mathcal{L}_{reg} = \frac{\langle \omega_1, \omega_2 \rangle^2}{|\omega_1|_2^2 |\omega_2|_2^2} = 0$ ensures that the two points will not be identical. We note that accuracy-optimum and fairness-optimum points becoming identical may not happen in experiments, we merely use this regularization term to prevent it from happening. Thus the final objective function will be $\mathcal{L} = \mathcal{L}_{ce} + \alpha\mathcal{L}_f + \beta\mathcal{L}_{reg}$ . The algorithm of YODO is presented in Algorithm 1.

**To Reviewer bMqY: Concern 4: weights to be orthogonal**

**To Reviewer zQ7a: Requested Change 3: analyze necessity of $\mathcal{L}_{reg}$.**

### 4.3 Prediction Procedure

After training a model $f(\mathbf{x}; (1 - \alpha)\omega_1 + \alpha\omega_2))$ with two sets of parameters $\omega_1$ and $\omega_2$, the prediction procedure for a test sample $\mathbf{x}$ can be summarized as follows:

1. Choose the desired trade-off parameter $\alpha$, which controls the balance between accuracy and fairness.
2. Compute the weighted combination of the two sets of trained weights, $(1 - \alpha)\omega_1 + \alpha\omega_2$, to obtain the model parameters for the desired trade-off.
3. Ccompute the prediction function to the test sample $\mathbf{x}$ as $f(\mathbf{x}; (1 - \alpha)\omega_1 + \alpha\omega_2)$, to obtain the predicted output.

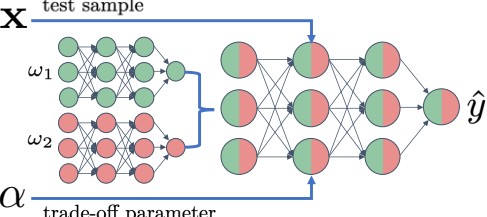

Figure 2: Prediction procedure of YODO

This prediction procedure offers flexible accuracy-fairness trade-offs at inference time, enabling users to choose the desired level of accuracy and fairness for their specific application.

**To Reviewer DMiX: Requested Changes 1: we added Section 4.3 for the prediction procedure.**

## 5 Discussion

In this section, we provide further analysis of our proposed method, including the model complexity, and model deployment. Additionally, we discuss the relationship between our work and related research.

**Model Complexity.** we analyze the time and space complexity of YODO. Compared to the memory utilization of the models with fixed accuracy-fairness trade-off, our method utilizes twofold memory usage. However, it can achieve in-situ flexible trade-offs at inference time. In practice, if the downstream task needs $N$ kinds of different accuracy-fairness trade-offs, we need $N$ different trained models with different fixed fairness constraints, which also need $N$ times training. In this sense, YODO only requires storing two sets of neural network weights and one-time training. For time complexity, the extra computational cost stems from the linear combination of the two sets of neural network weights at inference time, which is time-saving and negligible. At the training time, the computational cost is from gradient computation of $\omega_1/\omega_2$ (Equation (6)). We also conducted

Table 1: Comparing the running time for one epoch of the fixed model and the YODO, the experiments are conducted using an NVIDIA RTX A5000 GPU. The results are the mean of 10 trials The unit is second.

| Datasets | Fixed | YODO | Extra Time |
|---|---|---|---|
| UCI Adult | 0.42 | 0.58 | 38% |
| KDD Census | 3.78 | 4.99 | 32% |
| ACS-I | 3.93 | 6.09 | 55% |
| ACS-E | 3.53 | 4.25 | 20% |
| Average | 2.91 | 3.97 | 36% |

experiments comparing the fixed model and `YODO` to evaluate the additional running time and presented the results in Table 1. The results show that, on average, `YODO` only results in a 36% increase in training time. When compared to an arbitrary accuracy-fairness trade-off, this extra time is negligible. For example, if we require 100 levels of trade-off on the ACS-E dataset, `YODO` takes 2.91 seconds, while training 100 fixed models takes 397 seconds. In this sense, the 2.91 needed by `YODO` is considered negligible.

> **To Reviewer bMqY: Requested Change 3: add time complexity analysis**

**Model Deployment.** We discuss the advantages of the proposed method in the model deployment phase. `YODO` can be recalibrated for accuracy-fairness configuration before/after deployment for a fixed/flexible accuracy-fairness trade-off. For fixed accuracy-fairness trade-offs, we can set predefined $\alpha$ for `YODO` without retraining the model. For flexible accuracy-fairness trade-offs, we can define an API for users to input their $\alpha$ to meet their own requirements. Another advantage is that our method can output the varying predictive values for individuals (experiment in Figure 7), which can provide a reference for the decision makers to examine the realization degree of fairness.

**Difference and Relation to Previous Works.** Wortsman et al. (2021); Benton et al. (2021) proposed to learn the neural network subspaces, which aims to find a flat subspace for high accuracy. The goal of these two papers is to learn a *functionally diverse* neural network subspace in which the model accuracy is all high at each endpoint. Our method creatively proposes to achieve flexible fairness at inference time via learning a *objective-diverse* neural network subspace containing the optimum solutions for multiple objectives (e.g., accuracy and fairness). Nunez et al. (2023) proposed a similar method to achieve accuracy-efficiency trade-offs, which is similar to our proposed method. However, this paper proposed to train a degenerate subspace with a single point in weight space. Instead, we target to train an objective-diverse neural network subspace by training the linear combined weights with corresponding linear combinations of objectives (i.e., accuracy and fairness).

## 6 Experiments

We empirically evaluate the performance of our proposed method in this section. First, we evaluate the effectiveness of the proposed method on both tabular and image datasets in Section 6.1. Then, we investigate the in-situ flexible accuracy-fairness trade-offs of `YODO` in Section 6.2. Moreover, we investigate the visualization of hidden representation and the distribution of the prediction values in Sections 6.3 and 6.4, which gains insights into why our method can achieve flexible accuracy-fairness trade-offs. We also provide a case study on `CelebA` dataset in Section 6.5. The major **Obs**ervations of the experiments are highlighted in **boldface**.

**Experimental Setting.** In our experiments, we evaluate the performance of our proposed method on multiple datasets, including both tabular and image data. For tabular dataset, we adopt `UCI Adult` (Dua & Graff, 2017), `KDD Census` (Dua & Graff, 2017), `ACS-I` (ncome) (Ding et al., 2021), `ACS-E` (mployment) (Ding et al., 2021), which are widely used benchmark datasets in the fairness and machine learning community. For image data analysis, we use the `CelebA` dataset, which is a large-scale face attributes dataset with more than 200K celebrity images. This dataset is commonly used in the field of computer vision and has been used in various studies related to fairness in machine learning. The additional experimental setting is presented in Appendix A.

**Baselines.** We included two kinds of baseline methods, namely **Fixed Training** and **ERM**, in our experiments to validate the effectiveness of our proposed method. Fixed Training, as an important baseline, trains multiple models for different fixed accuracy-fairness trade-offs. Training the multiple fixed models requires expensive time and memory costs. For each point in Figures 3 and 12, we trained one fixed model using the objective function shown in Equation (2) with different values of $A$, which is set to $(0, 1]$ with an interval of 0.05. Besides that, we also consider more baseline methods, including Prejudice Remover (Kamishima et al., 2012), Adversarial Debiasing (Louppe et al., 2017) for demographic parity. [5] We provide more details of the baselines in Appendix B.2. For our experimental setting, we have to train 20 individual models from

> **To Reviewer zQ7a: Concern 6: Range of $A$ and $A = 0$.**

---

[5]We note that all these baselines use fixed training (i.e., each model represents a single level of fairness), while our proposed `YODO` trains once to achieve a flexible level of fairness.

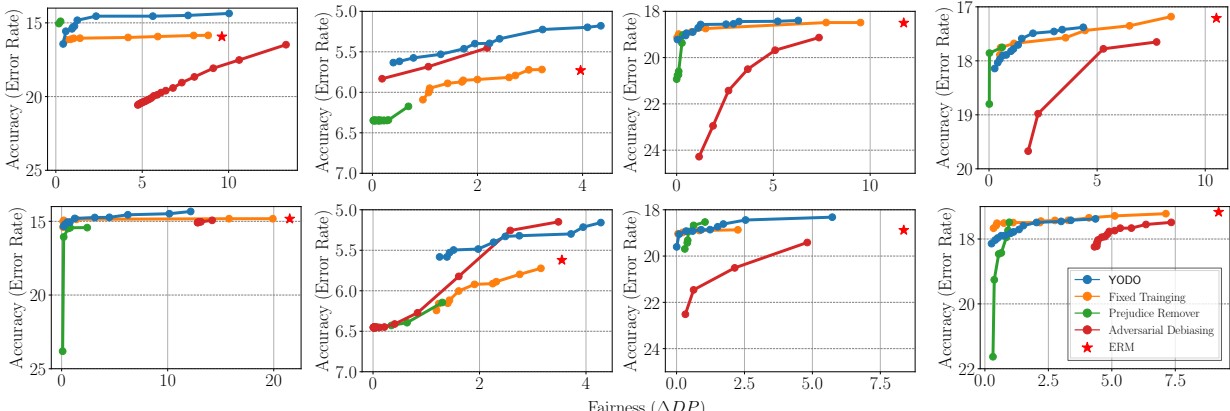

Figure 3: The Pareto frontier of the model accuracy and fairness. The first row is the fairness performance with respect to gender sensitive attribute, while the second row is race sensitive attribute. The model performance metric is Error Rate (lower is better), and the fairness metric is $\Delta DP$ (lower is better, Equation (1)).

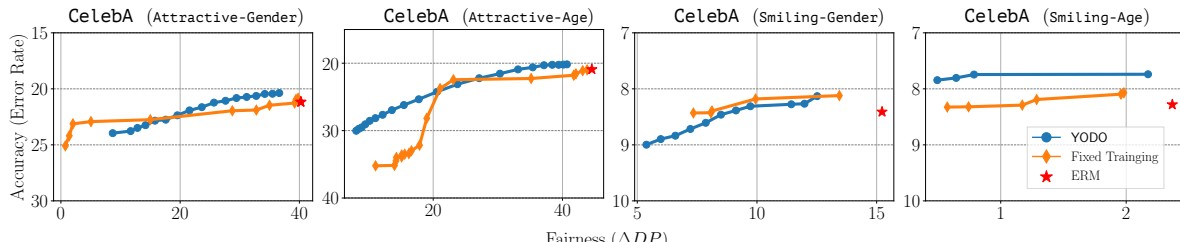

Figure 4: The Pareto frontier of the model performance and fairness on the `CelebA` dataset. The downstream task is to predict whether a person is Attractive (Smiling) or not. The sensitive attribute we considered is gender and age. The x-axis represents the difference in demographic parity ($\Delta$DP) between the sensitive groups, while the y-axis represents the error rate of the model. Our proposed one-time training model achieves a comparable trade-off between performance and fairness, as indicated by the Pareto frontier when compared to a set of fixed-trained models.

scratch for each trade-off hyperparameter. Another baseline is Empirical Risk Minimization (ERM), which is to minimize the empirical risk of downstream task (marked as ★ in the result).

### 6.1 Will YODO Achieve Flexible Trade-offs only with Training Once?

In this section, we validate the effectiveness of our proposed YODO on real-world datasets, including tabular data and image data. We present the results in Figures 3, 4 and 12 and use Pareto frontier (Emmerich & Deutz, 2018) to evaluate our proposed method and the baseline. Pareto frontier is widely used to evaluate the accuracy-fairness trade-offs by Kim et al. (2020); Liu & Vicente (2020); Wei & Niethammer (2020) and characterizes a model's achievable accuracy for given fairness conditions. Pareto frontier characterizes a model's achievable accuracy for given fairness conditions, as a measurement to understand the trade-offs between model accuracy and fairness. The Pareto frontier indicates the set of models that achieve the best trade-off between performance and fairness. The detail of the experiments can be found at Appendix B.4. The results on the tabular dataset are presented in Figures 3 and 12. And the results on the image dataset are presented in Figure 4. From these figures, we make the following major observations:

**Obs.1: Even though YODO only needs to be trained once, it performs similarly to baseline (Fixed Training), or even better in some cases.** After conducting experiments on real-world datasets, we compared the Pareto frontier of YODO with that of the **Fixed Training** baseline. We found that Pareto frontier of YODO coincides with that of **Fixed Training** in most figures, indicating it can achieve comparable or even better performance than baseline. The result makes YODO readily usable for real-world applications

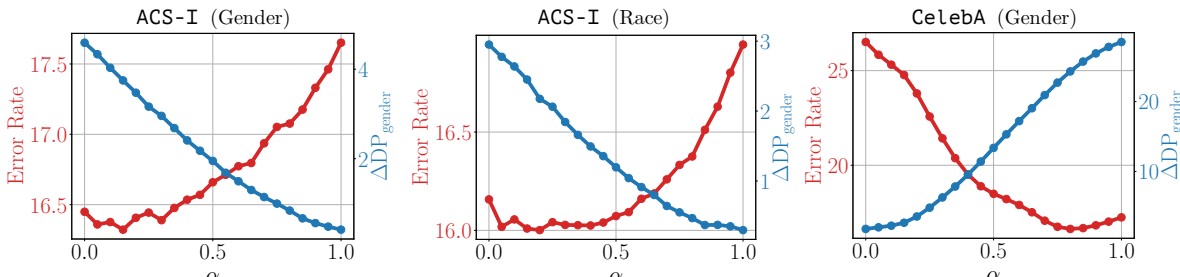

Figure 5: The accuracy-fairness trade-offs at inference time with respect to $\alpha$ for three different datasets: `ACS-I` dataset with gender as the sensitive attribute (**Left**), `ACS-I` dataset with race as the sensitive attribute (**Middle**), `CelebA` dataset with gender as the sensitive attribute (**Right**). We observed that the fine-grained accuracy-fairness trade-offs could be achieved by selecting different values of $\alpha$, providing more nuanced accuracy-fairness trade-offs. Note that the results are obtained at inference time with a single trained model.

requiring flexible fairness. It is worth highlighting that our proposed method only needs one-time training to obtain the Pareto frontier at inference time, making it computationally efficient.

**Obs.2:** YODO **achieves better fairness performance than baseline (Fixed Training) in some cases.** From the results of `UCI Adult`, `KDD Census`, and `ACS-I` dataset, we observed that our method achieves better fairness performance in some cases. Specifically, the Pareto frontier represented in blue in the figures can reach a lower $\Delta DP$ value than the baseline methods. This observation suggests that our method is highly effective in achieving flexible fairness and can be readily used in real-world applications where fairness is crucial for the downstream task.

**Obs.3: On some datasets (e.g., `UCI Adult`, `KDD Census`, `CelebA`),** YODO **even outperforms the ERM baseline.** Upon comparing our results on the `UCI Adult` and `KDD Census` datasets, we observed that the Pareto frontier of our method covers the point of ERM. This finding suggests that our approach outperforms ERM in both model accuracy and fairness performance. It also demonstrates that our proposed objective-diverse neural network subspace can achieve flexible accuracy-fairness trade-offs while potentially improving model accuracy. This advantage is likely attributed to the powerful capacity of the proposed objective-diverse subspace.

**Obs.4:** YODO **achieves a more smooth Pareto frontier than baseline (Fixed Training).** On most datasets, especially on `CelebA` image data, our proposed method demonstrates a smoother Pareto frontier compared to the baselines. For example, on `CelebA` Attractive-Age and Attractive-Gender in Figure 4, the Pareto frontier is notably smoother than baselines. A smooth Pareto frontier implies that our proposal is able to achieve more fine-grained accuracy-fairness trade-offs, which can provide a more dependable prediction for high-stakes decision-making scenarios.

### 6.2 Can the Accuracy-fairness Trade-Offs Be Flexible by Controlling Parameter $\alpha$?

To explore the effect of hyperparameter $\alpha$ on the accuracy-fairness trade-offs, we conducted experiments on the `ACS-I` and `CelebA` datasets with different values of $\alpha$. $\alpha$ is a hyperparameter that controls the trade-off between model accuracy and fairness in machine learning models. A higher value of $\alpha$ emphasizes more on fairness, while a lower value prioritizes accuracy. By adjusting the value of $\alpha$, we can make the accuracy-fairness trade-off flexible and controllable. We evaluate the model accuracy and fairness performance with the different values of $\alpha$ at inference time. The results are presented in Figure 5 and we observed:

**Obs.5: The accuracy-fairness trade-offs can be controlled by $\alpha$ at inference time.** Our experiments on both tabular and `CelebA` datasets showed that as we increase the value of hyperparameter $\alpha$, the error rate (lower is better) gradually increases while $\Delta DP$ gradually decreases. The solution at $\alpha = 0$ prioritizes accuracy-optimum (lowest error rate), while the solution at $\alpha = 1$ prioritizes fairness-optimum (lowest $\Delta DP$). These results demonstrate that our proposed method can achieve flexible accuracy-fairness trade-offs, with distinct linear correlations between the trade-off and the value of $\alpha$. Based on these findings, our approach can provide more reliable and controllable predictions by customizing $\alpha$ at inference time.

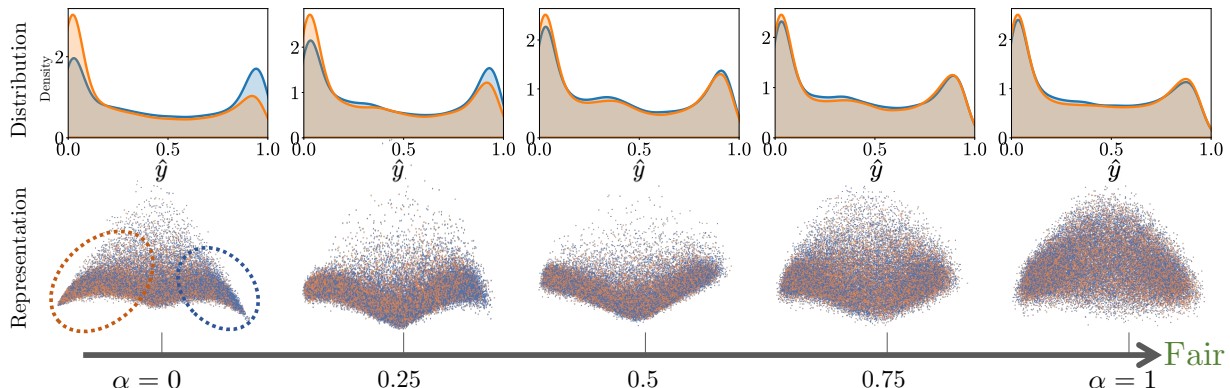

Figure 6: The changing distribution of the prediction values $\hat{y}$ and the representations as $\alpha$ increases. Blue indicates male and orange indicates female. **Top**: The distribution is estimated with kernel density estimation (Terrell & Scott, 1992). The distributions are more polarized between males and females with $\alpha = 0$, but become more similar with $\alpha = 1$. As $\alpha$ increases, the distributions for male and female groups become more similar, indicating achieving demographic parity. **Bottom**: The visualization of the representation with the values of $\alpha$ are from 0 to 1. The figure at the far left ($\alpha = 0$) shows that the representations for males ( ) and females ( ) are distinctly separate, indicating that the representations contain more sensitive information. The figure on the far right ($\alpha = 1$) shows the representations of different groups are mixed together, indicating that the representation contains little sensitive information.

### 6.3   How Does the Distribution of the Predictive Values Vary with Changing $\alpha$ ?

In this section, we examine the distribution of predictive values to investigate the varying trade-offs of our method. We specifically plot the distribution of predictive values for different groups (male and female) to verify the flexible accuracy-fairness trade-offs provided by our approach. The experiments are conducted on `ACS-I` dataset and the results are presented in Figure 6. We have the following observation:

**Obs.6: The distribution of predictive values for different groups becomes increasingly similar as the value of $\alpha$ increases**, indicating that our model becomes more fair as the values of $\alpha$ increase. Additionally, we found that the distributions of the predictive values of different groups follow the same distribution, showing the predictive values are independent of sensitive attributes. The varying distribution of predictive values provides valuable insights into why our model can achieve flexible accuracy-fairness trade-offs from a distributional perspective..

### 6.4   How Do the Representations Vary with Changing $\alpha$ ?

In this section, we visualize the hidden representation of different groups to provide further analysis and insight into the proposed method, YODO. Demographic parity requires that model predictions be independent of sensitive attributes (e.g., demographic groups). It is natural and rational to examine the independence between the hidden representation (penultimate layer) and the sensitive attribute. To interpret this representation, if $\alpha = 0$ (unfair model), the representations of the male and female groups would contain more sensitive information. In contrast, if $\alpha = 1$ (fair model), the representations would display a mix, signifying minimal sensitive information. The visualization of representations is widely used to inspect the fairness issue of DNNs from the manifold perspective (Du et al., 2021; Louizos et al., 2015). With the different combination with different $\alpha$s, we visualize the hidden representation in Figure 6 and we observed that:

To Reviewer bMqY: Concern 3: explain hidden representation.

**Obs.7: The disparity of the representation of different groups becomes smaller and smaller with the increasing $\alpha$.** The figure on the far left in Figure 6 shows that the representation of male and female groups are distinctly separate, indicating that the representations contain more sensitive information. The figure on the far right shows the representations of different groups mixed together, indicating that the representations contain little sensitive information. The sensitive information embedded in the representation

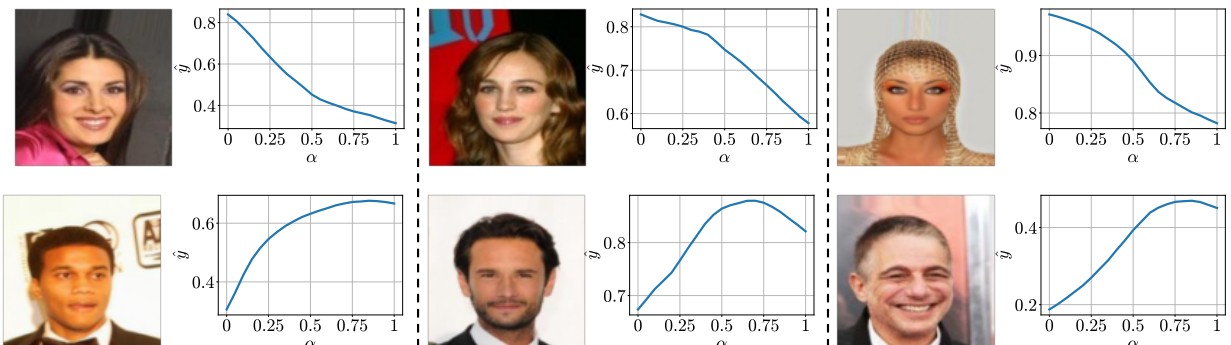

Figure 7: Case study on `CelebA` dataset. The sensitive attribute is gender. The downstream task is to predict whether a person is attractive or not. The results show that the `YODO` can provide the instance-level prediction change for practitioners to examine the fairness performance of our proposed method. By providing instance-level prediction changes, `YODO` enabled practitioners to better understand the model's fairness performance and make informed decisions.

indicates that the two endpoint biases correspond to model accuracy and fairness, respectively. This phenomenon provides insight into `YODO`, that it can learn a fair representation to guarantee a fair prediction.

### 6.5 How Does the Prediction Value Vary at the Instance Level? A Case Study on Image Data

In this section, we experiment on the `CelebA` dataset to investigate how the predicted values changes and the varying prediction with the various $\alpha$ in the instance level. The sensitive attribute we consider is Gender. The downstream task is to predict whether a person is attractive or not. The female group has more positive samples than the Male group, leading the biased prediction. We first present the mean of the prediction with respect to varying $\alpha$ for each group (i.e., male and female) in Figure 8. We also present some cases to investigate the effect of the change of the values of $\alpha$ in Figure 7. Based on our analysis of Figures 7 and 8, we make the following observations:

**Obs.8: The mean of the predictive values of different groups (e.g., female, male) approaches** 0.5 **with the increasing** $\alpha$. Analysis of Figure 7 shows that, overall, `YODO` tends to decrease the predictive values for the Female group and in-

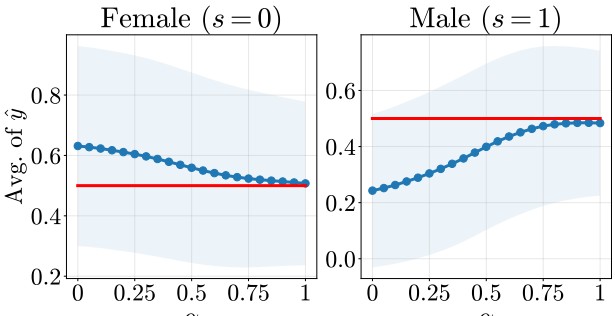

Figure 8: The mean of the prediction values for females and males with respect to varying $\alpha$. The red line indicates the predictive value $\hat{y} = 0.5$. The blue line is the mean of predictive values of each group (i.e., male and female). The results illustrate that the mean prediction values for males and females become more similar as $\alpha$ increases., indicating a fairer prediction.

crease the predictive values for the Male group, resulting in lower $\Delta DP$ values and indicating fairer predictions overall. Our observations demonstrate that our proposed method can mitigate the unfairness caused by the dataset's gender imbalance and ensure that individuals from different groups are treated equitably.

**Obs.9:** `YODO` **can provide an instance-level explanation for group fairness with only one model, while fixed training models need multiple models.** In `CelebA` dataset, the Female group has more positive (attractive) samples than the other one, leading to a higher predictive value for the male group. Figure 7 shows that `YODO` tends to lower the predictive values of the Female group while higher the predictive values of the Male group. Such a tendency would result in fairer predictive results. Our proposed method `YODO` offers an instance-level explanation for individuals with only one model . For example, Figure 7 demonstrates how `YODO` can lower the predictive values for the Female group and increase the predictive values for the Male group, resulting in a fairer outcome for individuals belonging to each group. Our method can provide individualized explanations and a trustworthy model for end-users.

To Reviewer bMqY: Requested Change 4: Rewrite observation 9.
To Reviewer bMqY: Requested Change 4

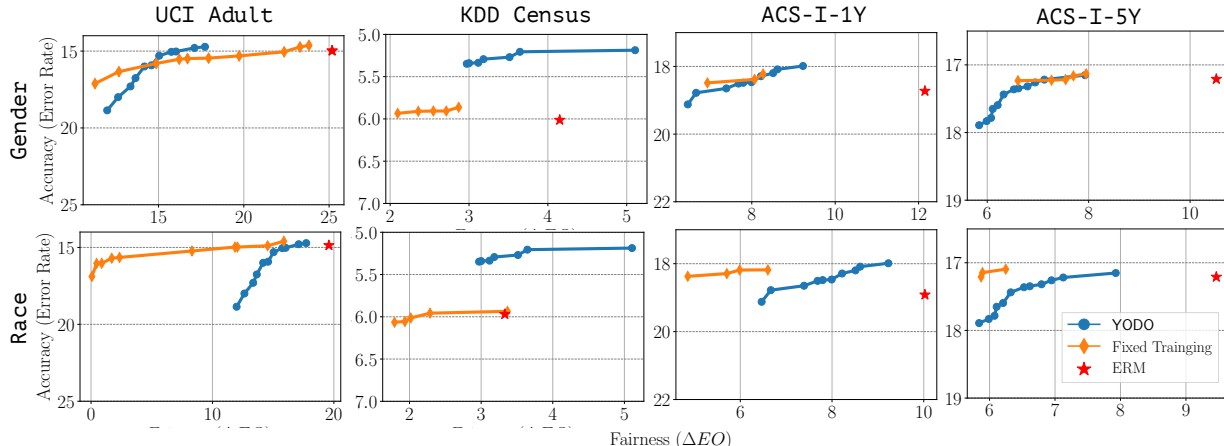

Figure 9: The Pareto frontier of the model accuracy and fairness. The first row is the fairness performance with respect to gender sensitive attribute, while the second row is race sensitive attribute. The model performance metric is Error Rate (lower is better), and the fairness metric is $\Delta EO$ (lower is better, Equation (1).

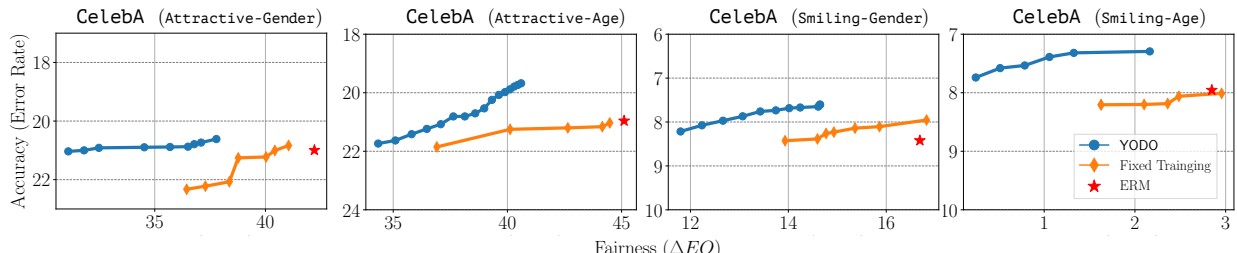

Figure 10: The Pareto frontier of the model performance and fairness on `CelebA` dataset. The downstream task is to predict whether a person is Attractive (Smiling) or not. The sensitive attribute is gender and age.

## 6.6 How Does YODO Perform on Another Group Fairness Criterion, Equality of Opportunity?

In this section, we experiment on the fairness metric Equality of Opportunity (EO) (Hardt et al., 2016). EO measures whether a classifier provides equal opportunities to individuals from different groups. By satisfying this criterion, we can ensure that protected groups are not disproportionately negatively impacted by the model's predictions.

**Definition 6.1** (Equality of opportunity). *A classifier satisfies this definition if both protected and unprotected groups have an equal probability of a subject in a positive class having a negative predictive value.*

For example, this implies that the probability of an applicant with an actual good credit score being incorrectly assigned a bad predicted credit score should be the same for both male and female applicants. The formal definition of EO is as the following:

$$\Delta \text{EO}(f) = \left| \mathbb{E}_{\mathbf{x} \sim \mathcal{D}_0, y=1} f(\mathbf{x}) - \mathbb{E}_{\mathbf{x} \sim \mathcal{D}_1, y=1} f(\mathbf{x}) \right|. \tag{7}$$

We conduct the experiment using $\Delta EO$ as a fairness constraint and metric and present the results on tabular and image data. The results are presented in Figures 9, 10 and 13. We have the following observations:

**Obs.10:** YODO **performs similarly to baseline (Fixed Training) in terms of Equality of Opportunity, or even better in some cases.** This observation indicates it can achieve comparable or even better performance than the baseline. And this experiment also makes our method readily usable for real-world applications, which require flexible fairness. The result shows the effectiveness of our proposal on other fairness metrics, demonstrating its overall effectiveness in promoting fairness.

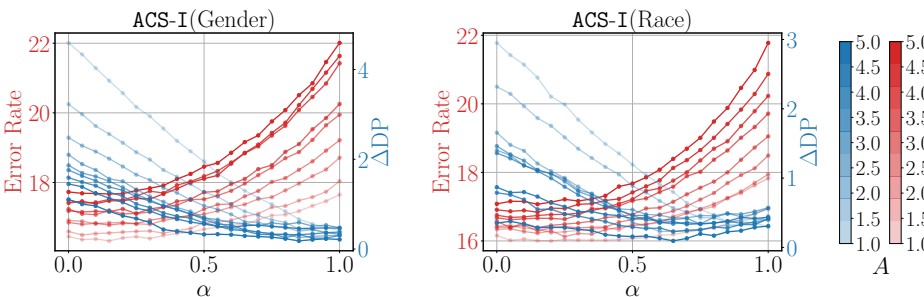

Figure 11: The effect of the accuracy-fairness balance parameter $A$. We train YODO with varying training balance parameter values ($A$) and report their performance with different $\alpha$. The x-axis represents a parameter to control the accuracy-fairness trade-off for inference, while the y-axis shows both fairness (in blue) and accuracy (in red). The strength of the color reflects the value of $A$. The optimal balance, achieving the best performance in terms of accuracy and fairness, occurs at $A = 1$. We also present the results for each figure corresponding to different values of $A$ in Figures 17 and 18.

### 6.7 How Does YODO Perform with Different the Balance Parameter $A$?

In this experiment, we evaluated the performance of the *Yodo* model with varying balance parameter values ($A$). The desirable balance parameter $A$ for the accuracy-fairness trade-off should have the following properties:

1. The error rate of the downstream task should be as low as possible.

2. The best performance for fairness should be achieved when we choose the hyperparameter $\alpha$ at inference time.

To investigate the impact of the value $A$ on the trade-off between accuracy and fairness, we tested the model performance with varying $A$, ranging from 1 to 5, with increments of 0.5. We present the results in Figure 11.

The results show that YODO exhibited its best performance when $A = 1$, as this specific balance parameter value resulted in higher accuracy and a larger demographic parity span compared to other values of $A$. This observation indicates that setting $A = 1$ effectively addresses the trade-off between model accuracy and fairness, achieving an optimal balance.

> To Reviewer 21XQ: Concern 4: varying $A$.

As the value of $A$ increases, although the fairness performance improves, the accuracy of the downstream task deteriorates. When $A = 5$, the error rate of the downstream task is even worse than the highest error rate of the model with $A = 1$. The poorer performance for the downstream task demonstrates that the model with $A = 5$ is inferior to that with $A = 1$.

> To Reviewer bMqY: Concern 1: varying $A$.

## 7   Conclusion

In this paper, we proposed YODO, a novel method that achieves accuracy-fairness trade-offs at inference time to meet the diverse requirements of fairness in real-world applications. Our approach is the first to achieve flexible trade-offs between model accuracy and fairness through the use of an objective-diverse neural network subspace. Our extensive experiments demonstrate the effectiveness and practical value of the proposed approach, and we offer a detailed analysis of the underlying mechanisms by examining the distributions of predictive values and hidden representations. By enabling in-situ flexibility, our approach can provide more nuanced control over the trade-offs between accuracy and fairness, thereby advancing the state-of-the-art in the field of algorithmic fairness.

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

In this appendix, we provide the two additional experiments in Appendix A, including the experiment on `ACS-E` dataset and the distribution of the predictive values on `CelebA` dataset. We also provide a more detailed experimental setting in Appendix B. We carried out supplementary experiments to delve deeper into the analysis of our proposed model in Appendix C.

## A  Additional Experiments

### A.1  Experiments on `ACS-E` Dataset

In this appendix, we present the results of complementary experiments conducted on the `ACS-E` dataset with gender as the sensitive attribute. The results for two-group fairness metrics, $\Delta$DP and $\Delta$EO, are presented in Figures 12 and 13, respectively. These results are complementary to the ones presented in Figures 3 and 9.

The experimental results on the `ACS-E` dataset with gender as the sensitive attribute show that our proposed method achieves comparable accuracy-fairness trade-offs to the baselines for both group fairness metrics $\Delta$DP and $\Delta$EO. This finding is consistent with the results of the experiments on other datasets with different sensitive attributes presented in Figures 3 and 9.

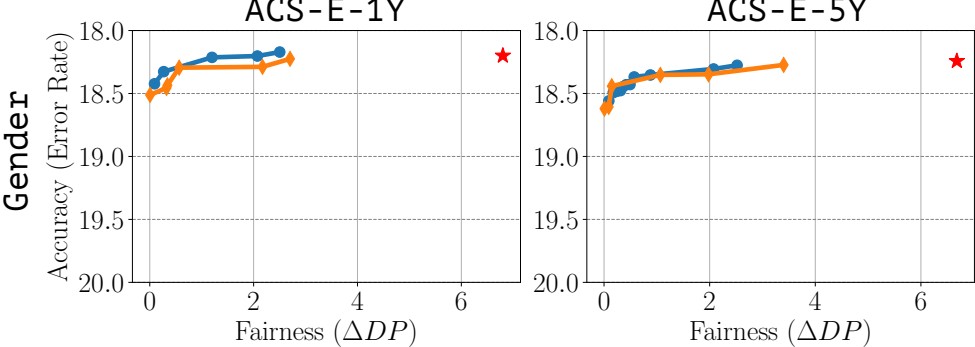

Figure 12: The Pareto frontier of the model performance and fairness on `ACS-E` dataset spanning 1 year or 5 years on $\Delta DP$ metric. The sensitive attribute is gender.

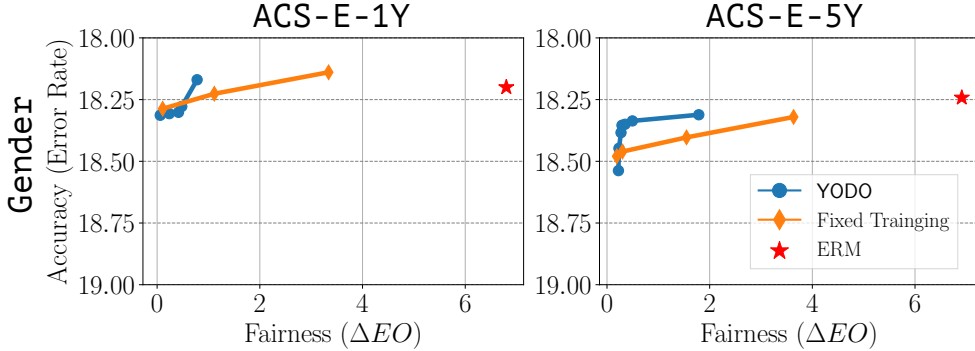

Figure 13: The Pareto frontier of the model performance and fairness on `ACS-E` data spanning 1 year or 5 years on $\Delta EO$ metric. The sensitive attribute is gender.

### A.2  The Distribution of Predictive Values on Image Dataset

In this appendix, we provide the distribution of the predictive values on the image dataset, `CelebA`. The distribution of the predictive values of different groups are more and more similar with the increase of the values of $\alpha$, indicating that our model is more and more fair with the increase of the values of $\alpha$. This result shows that our proposed method encourages the distribution to follow the same distribution, and further guarantees fair prediction.

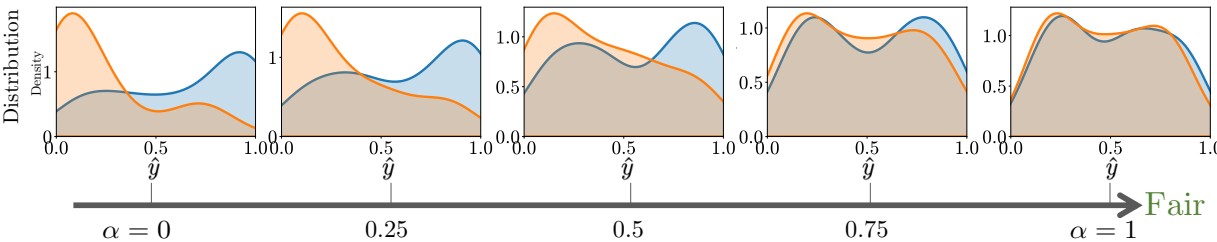

Figure 14: The distribution of the prediction values with different $\alpha$ on `CelebA` dataset. The distribution of the predictive values of different groups (i.e., Male, Female) becomes more and more similar with the increasing $\alpha$. The distributions are more polarised between Male and Female with $\lambda = 0$ while the distribution is nearly the same with $\lambda = 1$.

Table 2: The summary of the datasets used in our experiment.

| Dataset | Data Type | Task | Sensitive Attributes | #Instances |
|---|---|---|---|---|
| `UCI Adult` | Tabular | Income | Gender, Race | 48,842 |
| `KDD Census` | Tabular | Income | Gender, Race | 299,285 |
| `ACS-Income` (1 year) | Tabular | Income | Gender, Race | 265,171 |
| `ACS-Income` (5 years) | Tabular | Income | Gender, Race | 1,315,945 |
| `ACS-Emploement` (1 year) | Tabular | Employment | Gender | 136,965 |
| `ACS-Emploement` (5 years) | Tabular | Employment | Gender | 665,137 |
| `CelebA` | Image | Attractive | Gender, Age | 202,599 |

# B    Experiment Details

In this section, we provide a detailed description of our experiment setting, including the description of the datasets, neural network architectures, and experiment settings used in our experiments.

## B.1    Dataset

In the appendix, we provide more details about the datasets used in the experiments. These include tabular datasets such as `UCI Adult`, `KDD Census`, `ACS-I`, `ACS-E`, as well as the image dataset `CelebA`. We provide the statistics of the datasets in Table 2. In the following, we provide a comprehensive description of each dataset used in our experiments:

- **`UCI Adult`**[6] (Dua & Graff, 2017): This dataset is extracted from the 1994 Census database. The downstream task is to predict whether the personal income is over 50K a year. The sensitive attributes in this dataset are gender and race.

- **`KDD Census`**[7] (Dua & Graff, 2017): This data set contains 299285 census data extracted from the 1994 and 1995 by the U.S. Census Bureau. The data contains 41 demographic and employment related variables. The instance weight indicates the number of people in the population that each record represents due to stratified sampling. The sensitive attributes are gender and race.

- **`ACS-I`**(ncome)[8] (Ding et al., 2021): The task is to predict whether the income of an individual is above $50,000$. The source data was filtered to only include individuals above the age of 16, who reported usual working hours of at least 1 hour per week in the past year, and an income of at least $100$. The threshold of the income is $50,000$. We use two data from this dataset which spans 1 year or 5 years. The sensitive attributes we consider for this dataset are gender and race.

---

[6]https://archive.ics.uci.edu/ml/datasets/adult
[7]https://archive.ics.uci.edu/ml/datasets/Census-Income+(KDD)
[8]https://github.com/zykls/folktables

- **ACS-E**(mployment)[9] (Ding et al., 2021): The task is to predict whether an individual is employed and the individuals are between the ages of 16 and 90. We use two data from this dataset which spans 1 year or 5 years. The sensitive attribute we consider for this dataset is gender.

- **CelebA**[10] (Liu et al., 2015): The CelebFaces Attributes (**CelebA**) dataset a large-scale face attributes dataset consisting of more than 200K celebrity images and each image has 40 face attributes. The downstream task is to predict whether the person is attractive (smiling) or not, formulated as binary classification tasks. We consider Male (gender) and Young (age) as sensitive attributes.

## B.2   Baselines

We provide the details of the baseline methods employed in our experiments. We note that all baselines use fixed training (i.e., each model represents a single level of fairness), whereas our proposed YODO trains once to achieve a flexible level of fairness. The details of the baseline methods are as follows:

> **To Reviewer DMiX: Requested change 2: adding baselines.**

- **Fixed Training** (Dua & Graff, 2017) is an in-process technique that incorporates fairness constraints as regularization term into the objective function (Chuang & Mroueh, 2020; Kamishima et al., 2012). This approach enhances the model's fairness by optimizing the regularization term during training. The regularization term is represented as $\Delta DP$, $\Delta EO$, and $\Delta Eodd$, as shown in Equations (1), (7) and (8).

- **Prejudice Remover** (Kamishima et al., 2012) introduces the prejudice remover as a regularization term, ensuring independence between the prediction and the sensitive attribute. Prejudice Remover uses mutual information to quantify the relationship between the sensitive attribute and the prediction, thereby maintaining their independence.

- **Adversarial Debiasing** (Louppe et al., 2017) involves simultaneous training of the network for the downstream tasks and an adversarial network. The adversarial network receives the classifier's output as input and is trained to differentiate between sensitive attribute groups in the output. The classifier is trained to make accurate predictions for the input data while also training the adversarial network not to identify groups based on the classifier's output.

## B.3   Neural Network Architectures.

The experiments were conducted using a Multilayer Perceptron (MLP) neural network architecture for tabular data and a ResNet-18 architecture for image data.

- **Tabular**. Tabular data is structured data with a fixed number of input features. For our experiments, we used a two-layer Multilayer Perceptron (MLP) with 256 hidden neurons. The MLP architecture is commonly used for tabular data.

- **Image**. We used a ResNet-18 architecture for image data in our experiments. The used ResNet-18 has been pre-trained on the ImageNet dataset. And we fine-tuned it for our task. The ResNet-18 architecture is widely used for image classification tasks, as it can handle the high dimensionality of image data and learn hierarchical representations of the input features.

## B.4   Experiment setting

In our experiments, we trained the neural network using the Adam optimizer (Kingma & Ba, 2014). The optimization process took into account two important metrics: accuracy and fairness. To maintain focus on these metrics, we trained the neural network for fixed numbers of epochs across different datasets. Specifically,

---

[9] https://github.com/zykls/folktables
[10] https://mmlab.ie.cuhk.edu.hk/projects/CelebA.html

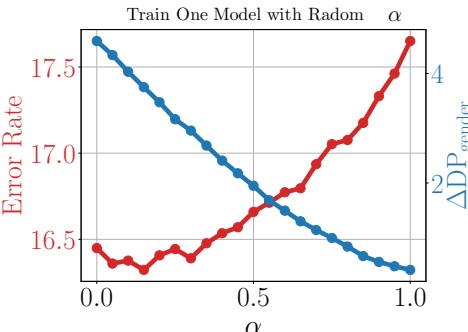 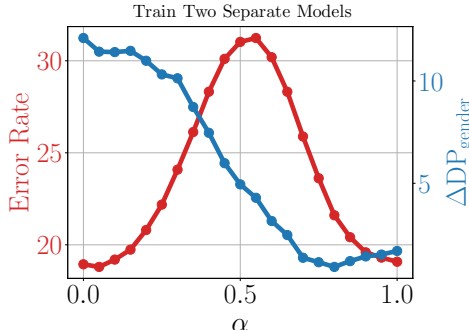

Figure 15: Comparison of Two Training Strategies. **Left:** Training a single model with two sets of parameters $\omega_1$ and $\omega_2$. **Right:** Training two separate models with $\omega_1$ and $\omega_2$. The error rate in the right subfigure shows that interpolating the weights $\omega_1$ and $\omega_2$ of the two well-trained models at inference time achieves no worse accuracy, since the error rate is relatively high when $\alpha = 0.5$.

for the `UCI Adult` and `KDD Census` datasets, we trained for 2 epochs, whereas the `ACS-I` and `ACS-E` datasets required 8 epochs of training. The `CelebA` dataset, on the other hand, demanded 30 epochs for adequate performance. We initialized the neural networks using the Xavier initialization (Glorot & Bengio, 2010). As for the training parameters, we set the learning rate to 0.001. We also used different batch sizes for the training process, with 512 being the batch size for tabular datasets, and 128 for image datasets. This difference in batch size accommodates the varying computational requirements of the datasets. Notably, we did not apply weight decay in our experiments.

**To Reviewer 21XQ: Requested Change 2: Fully described experimental details.**

## C  Additional Experiments

We carried out supplementary experiments to delve deeper into the analysis of our proposed model. These experiments encompass the following aspects: comparing the training of two separate models, examining Equalized Odds, investigating the impact of varying the accuracy-fairness balance parameter $A$, and evaluating the performance of the YOLO model with larger architectures.

### C.1  Comparison to Training Two Separate Models

Training two models separately and interpolating them can not achieve our goal- flexible accuracy-fairness trade-offs at inference time. As verified and studied by previous works Frankle et al. (2020); Wortsman et al. (2021); Fort et al. (2020); Benton et al. (2021), interpolating the weights $\omega_1$ and $\omega_2$ of two well-trained models at inference time has been shown to achieve no better accuracy than an untrained model. We also conduct experiments to verify this point in our case with two settings: 1) training a single model with two sets of parameters $\omega_1$ and $\omega_2$. 2) Training two separate models with $\omega_1$ and $\omega_2$. We present the results in Figure 15. The error rate in the right subfigure shows that interpolating the weights $\omega_1$ and $\omega_2$ of the two well-trained models at inference time achieves no worse accuracy since the error rate is relatively high when $\alpha = 0.5$.

### C.2  Experiments on Equalized Odds

In this section, we experiment on the fairness metric Equalized Odds (Eodd) (Hardt et al., 2016). Eodd assesses whether a classifier offers equal opportunities to individuals from diverse groups. A classifier adheres to this criterion if it maintains equal true positive rates and false positive rates across all demographic groups. We present a relaxed version of Eodd as follows:

$$\Delta \mathrm{Eodd}(f) = |\mathbb{E}_{\mathbf{x}\sim\mathcal{D}_0, y=1} f(\mathbf{x}) - \mathbb{E}_{\mathbf{x}\sim\mathcal{D}_1, y=1} f(\mathbf{x})| + |\mathbb{E}_{\mathbf{x}\sim\mathcal{D}_0, y=0} f(\mathbf{x}) - \mathbb{E}_{\mathbf{x}\sim\mathcal{D}_1, y=0} f(\mathbf{x})|. \tag{8}$$

We observed that YODO **performs similarly to baseline (Fixed Training) in terms of Equalized Odds, or even better in some cases.** This observation indicates our proposed method can achieve

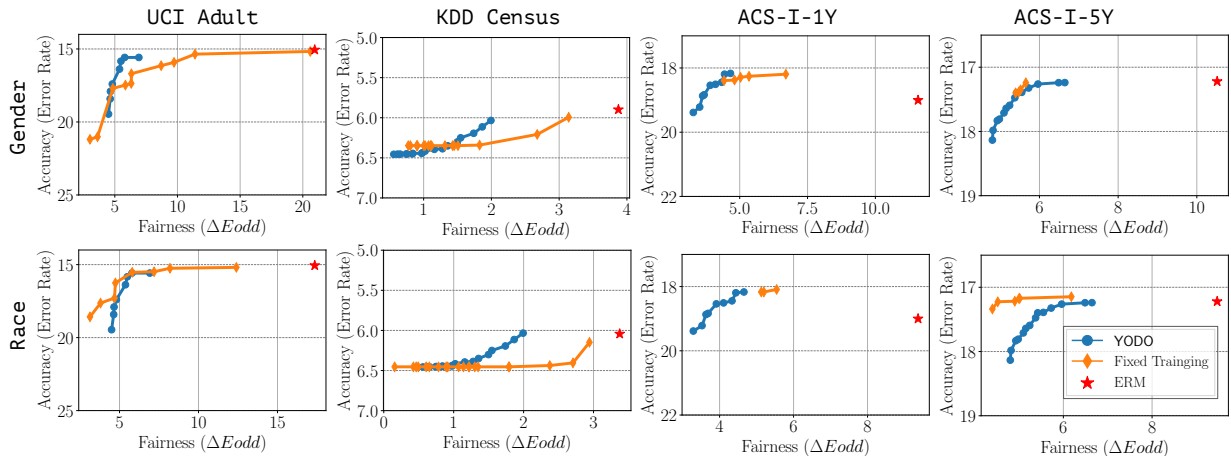

Figure 16: The Pareto frontier of the model accuracy and fairness. The first row is the fairness performance with respect to gender sensitive attribute, while the second row is race sensitive attribute. The model performance metric is Error Rate (lower is better), and the fairness metric is $\Delta Eodd$ (lower is better, as shown in Equation (1)).

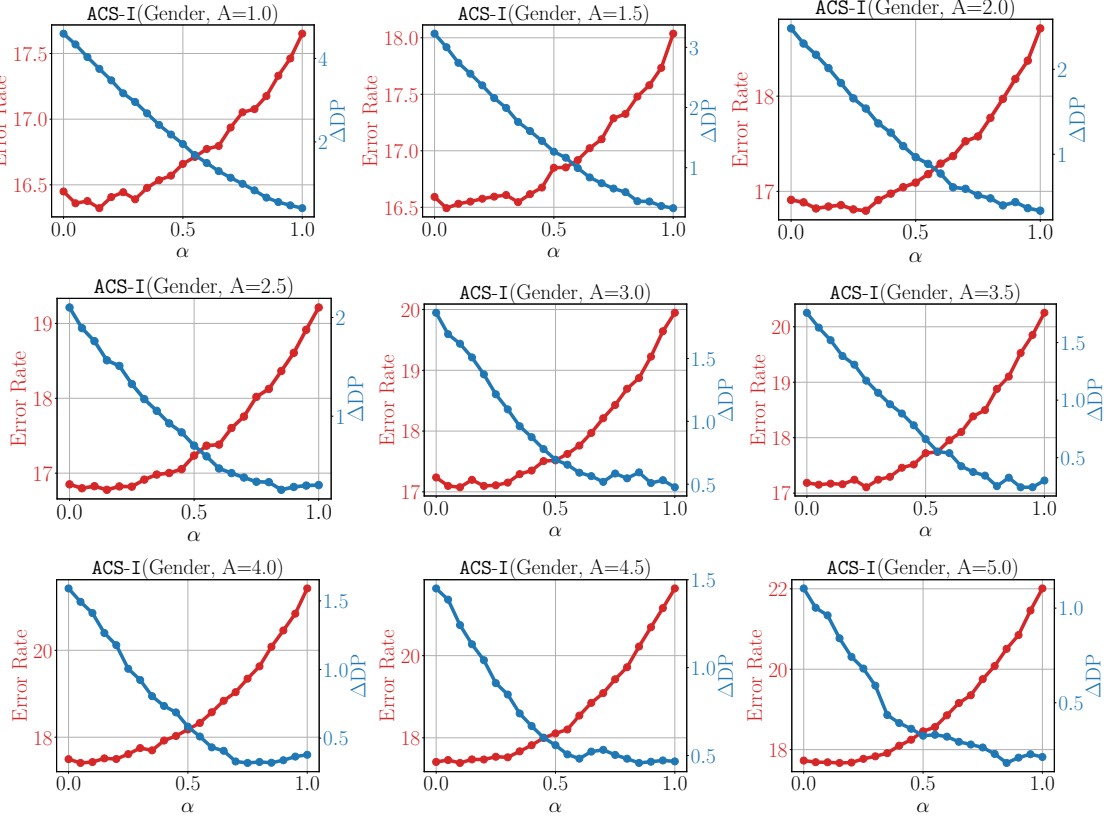

Figure 17: The effect of the accuracy-fairness balance parameter $A$ on ACS-I dataset with Gender as the sensitive attribute. The results show that our method can achieve flexible accuracy-fairness trade-offs at inference time with all the values of $A$.

comparable or even better performance than the baseline. The result shows the effectiveness of our proposal on other fairness metrics, demonstrating its overall effectiveness in promoting fairness.

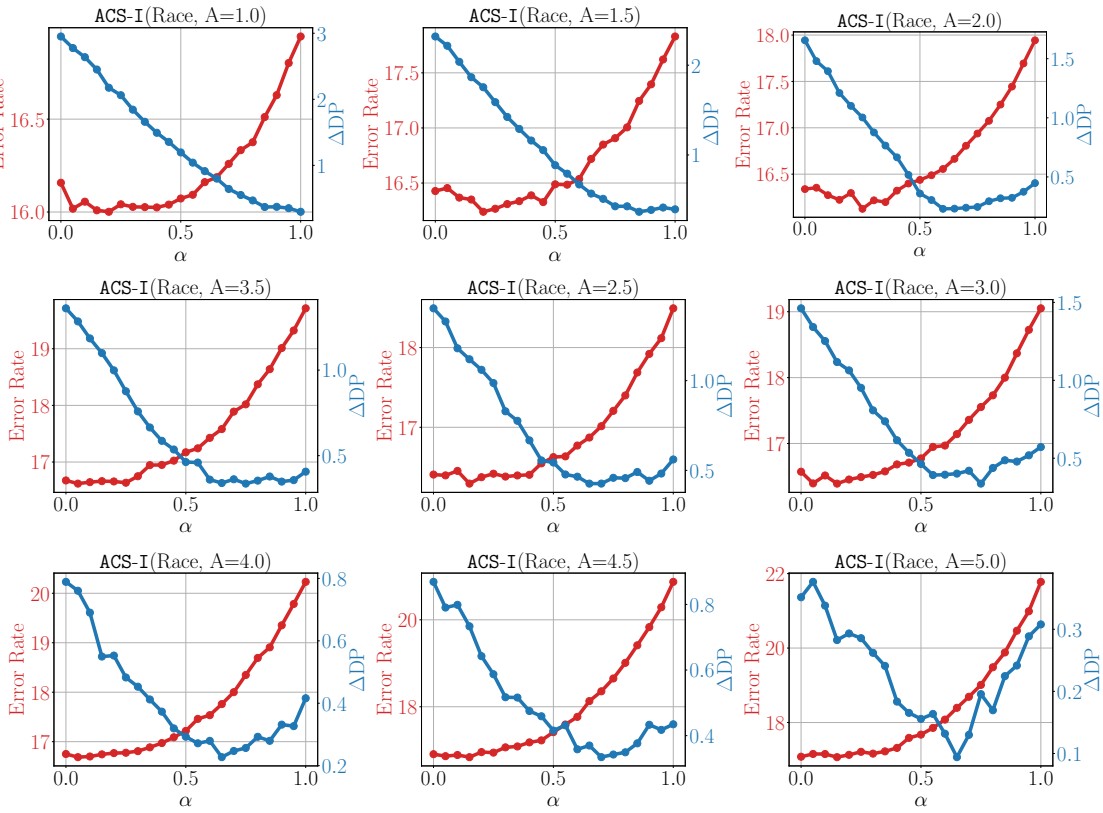

Figure 18: The effect of the accuracy-fairness balance parameter $A$ on `ACS-I` dataset with Race as the sensitive attribute. The results show that our method can achieve flexible accuracy-fairness trade-offs at inference time with all the values of $A$.

### C.3 Impact of Different Values of Accuracy-fairness Balance Parameter $A$

In addition to the combined results for different $A$ values presented in Figure 11, we also display the results for each individual $A$ value in separate figures, as shown in Figure 17 and Figure 18. The results demonstrate that models with various $A$ values can achieve a range of accuracy-fairness trade-offs. However, as $A$ increases, the overall accuracy of the downstream task declines while the span of the fairness metric $\Delta DP$ expands.

### C.4 The Performance of YODO with Larger Models

To Reviewer zQ7a: Concern 2: larger models.

We conducted experiments on larger models using the `CelebA` dataset, specifically examining ResNet18, ResNet34, ResNet50, ResNet101, WideResNet50, and WideResNet101 (He et al., 2016; Zagoruyko & Komodakis, 2016). We investigated the accuracy-fairness trade-off behavior of these models while varying the $\alpha$ parameter. We note that all the results are referenced with one model at the inference time. Our observations revealed that 1) All models were capable of achieving a flexible accuracy-fairness trade-off during inference. 2) Larger models demonstrated greater stability, with smaller models like ResNet18 and ResNet34 exhibiting more fluctuations, while larger models such as ResNet50, ResNet101, WideResNet50, and WideResNet101 showed smoother performance.

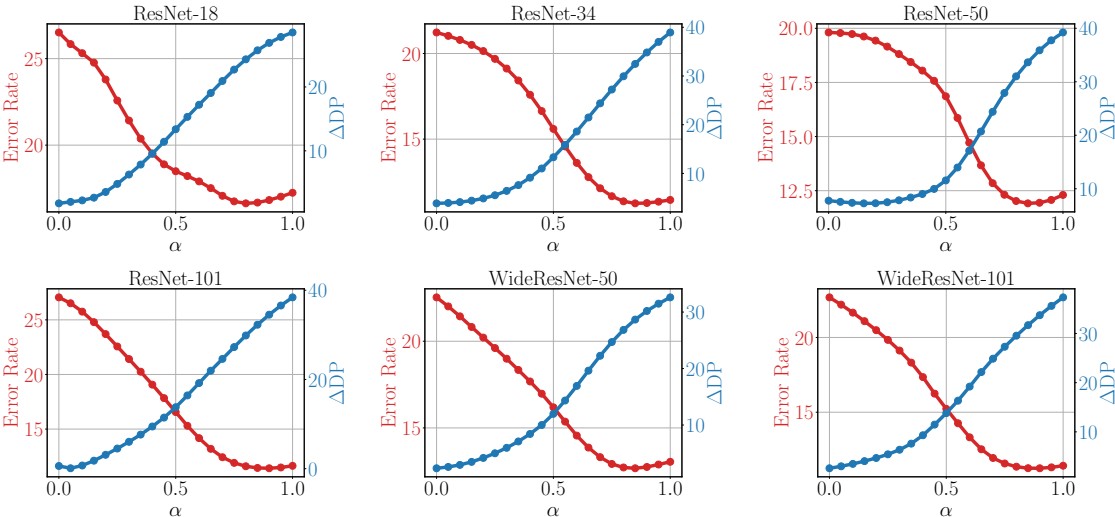

Figure 19: Performance of YODO with larger meodels.

