# OpenReview forum: "You Only Debias Once: Towards Flexible Accuracy-Fairness Trade-offs at Inference Time"
_TMLR — Rejected by TMLR_

### Review · Reviewer_bMqY · 2023-03-26

**Summary Of Contributions:**

In many scenarios, machine learning models need to find a trade-off between the accuracy of the predictions, and the fairness of these (given a fairness measure). This trade-off is usually fixed beforehand via a hyperparameter $\alpha$, and providing different trade-offs requires re-training the network from scratch. This work proposes to train a single neural network which can set this trade-off level $\alpha$ dynamically. In particular, the authors propose to parametrize the network as a linear combination of two different set of weights, $\alpha w_1 + (1 - \alpha) w_2$, in which this $\alpha$ parameter is tied to the one of the objectives trade-off $\alpha L_{acc} + (1-\alpha)(L_{acc} + L_{fair})$, such that the linear combination of weights optimizes the same linear combination of objectives. Then, the method is empirically tested in different tabular and image datasets, showing different aspects of this trade-off such as raw metrics, $\alpha$-traversals, and the learned hidden representations.

**Audience:**

Yes

**Broader Impact Concerns:**

Given that this is a work on the field of *fairness*, I find rather concerning that one of the experiments (CelebA) has as objective measuring how *attractive* a person is. Even if you could do this fairly (independently of the gender), having a machine learning model to predict attractiveness is rather unsettling and morally questionable.

**Claims And Evidence:**

No

**Requested Changes:**

**Necessary changes**

- **Writing.** While the writing could be improved a bit (e.g., "the inference time" -> "inference time"), I'd urge the authors to make sure that the method description and the mathematics are correct throughout the paper. For instance, I am pretty sure that the analysis of $w_1$ and $w_2$ in Sec. 3.2 is backwards.
- **Literature review and model comparisons.** I find remarkable that there is no mention of similar methods and thereof a lack of comparisons with them. While there are a number of batch citations in broad fields in Sec. 6, this work completely omits any prior work on finding this trade-off. Some relevant works that come to mind:
    - [You Only Train Once: Loss-Conditional Training of Deep Networks](https://openreview.net/forum?id=HyxY6JHKwr). I am *really* surprised that this work went missing, given the similarity of the titles. In a nutshell, the paper proposes an extremely similar approach as the submitted work, but the network weights are parametrized by an MLP hypernetwork, instead of by a linear combination of weights.
    - [Hypernetworks](https://arxiv.org/abs/1609.09106). A hypernetwork is a network whose output is the parameters of another network. In its simplest form, this hypernetwork is a linear combination of weights, which is essentially what this paper uses.
    - Multi-objective optimization. This one I am also surprises is missing, given that the authors talk about Pareto frontiers. Multi-objective optimization is no stranger to Machine Learning, and there are two main types of MOO: a priori, where the trade-off is fixed beforehand; and a posteriori, where the trade-off is set by the practitioner afterwards. The latter is essentially the objective of this work, and there is a bulk of work on a-posteriori MOO in Machine Learning, in which accuracy-fairness trade-off is a common application. See, for example, [Learning the Pareto Front with Hypernetworks](https://arxiv.org/abs/2010.04104) and [Efficient Multi-Objective Optimization for Deep Learning](http://arxiv.org/abs/2103.13392).
- **Time complexity** The authors should properly address the extra time complexity of the proposed method. The proposed formulation includes another expectation with respect to $\alpha \sim U$, which implies extra overhead. However, this average is *approximated* by taking a single sample per batch.

**Desirable changes**

- **D1** Rewrite observation 9, as providing instance-level explanation is not a contribution of this work. The contribution is not having to train several models to achieve this, which was already discussed in previous paragraphs.
- **D2** Please move the literature review earlier in the manuscript.

**Questions**

- **Q1.** Why do you train the "fair weights" on the regularized objective with $A=1$. It seems rather arbitrary, and it you want to be as fair as possible, I would at least increase $A$ quite a bit.
- **Q2.** Are the plots in Figure 2 and 3 correct? It seems rather odd that (to my understanding) there is no accuracy-fairness trade-off there at all: the most accurate models are always the fairest ones. Pareto fronts of this kind usually look like a concave curve, just like the ones of the MOO papers I referenced above. **Edit:** Reading my peers' reviews, I realized that the y-axis is inverted.

**Strengths And Weaknesses:**

**Strengths**
- The motivation of this work is clear, and of importance to the community and practitioners.
- The proposed idea is simple, yet powerful.
- The paper is self-contained, and well-written for the most part.

**Weaknesses**
- There are a few typos in the manuscript that can confuse the reader.
- The paper makes underlying assumptions (e.g. that a trade-off between accuracy and fairness is needed), which would justify some design choices like the regularizer $L_{reg}$, but at the moment are not justified. For this particular example, why would I want the model weights to be orthogonal? What if the model could find the best accuracy while being fair?
- There are no comparisons with any other existing method, and the literature review is rather uninformative.
- While the model can be really useful, its quantitative performance is not as impressive and, combined with the previous item, shades the potential of the proposed work.
- Some aspects of the paper are unclear to me. For example, what hidden representation are we measuring exactly in Fig. 5.

---

> ### Author Response · Authors · 2023-04-10
> **Response to Reviewer bMqY [1/3]**
>
> Dear Reviewer bMqY,
>
> We appreciate the time you have taken to review our paper and provide valuable feedback. We are pleased to hear that you found the motivation clear, our proposed idea powerful, and our paper well-written. We have carefully addressed your requested changes and concerns to improve the paper, and we have also revised the paper according to your concerns, labeled in ${\color{blue} blue}$. The detailed responses are as follows:
>
>
>
> ### Requested Changes
> **1. Writing. While the writing could be improved a bit (e.g., "the inference time" -> "inference time"), I'd urge the authors to make sure that the method description and the mathematics are correct throughout the paper. For instance, I am pretty sure that the analysis of $w_1$ and $w_2$ in Sec. 3.2 is backwards.** \
> We thank you for the suggestion for the writing and your careful reading of our paper. We revise "the inference time" to "inference time" throughout the entire paper.
>
> We truly appreciate you for pointing out that the order of $w_1$ and $w_2$ in Section 3.2 was backwards. We have thoroughly reviewed and revised the entire paper to ensure consistency throughout the paper.
>
> In addition to this, we carefully revise the entire paper to fix the grammar error.
>
>
> **2. Literature review and model comparisons.** \
> We thank you for your great suggestions!
>
> **Literature review** We have included the paper you mentioned regarding hypernetworks for loss conditional training. Additionally, we discussed multi-objective optimization in the context of Pareto frontiers (with a focus on accuracy and fairness in our paper).
>
> **Model comparisons** In order to further substantiate the effectiveness of our proposed method, we integrated widely-adopted baselines for demographic parity, such as 1) Prejudice Remover and 2) Adversarial Debiasing, and showcased the updated results in Figure 3. The outcomes continue to demonstrate the efficacy of our approach.
>
> It is crucial to clarify that our primary contribution is the ability of our method to achieve flexible accuracy-fairness trade-offs at inference time. As a result, our approach complements existing fairness methods rather than competing with them. Furthermore, our method can be seamlessly integrated into existing fairness techniques.
>
>
>
> **3. Time complexity The authors should properly address the extra time complexity of the proposed method. The proposed formulation includes another expectation with respect to , which implies extra overhead. However, this average is approximated by taking a single sample per batch.** \
> We thank you for your valuable comments. To further analyze the time complexity of our proposed method. We added the experiment to compare the fixed model and YODO to evaluate the additional running time and the results are in the following table. We report the running time for one epoch of the fixed model and the YODO, the experiments are conducted using an NVIDIA RTX A5000 GPU. The results presented are the mean of $10$ trials The unit is second.
>
> | Datasets   | Fixed | YOLO | Extra Time |
> |------------|-------|------|------------|
> | UCI Adult  | 0.42  | 0.58 | 38%        |
> | KDD Census | 3.78  | 4.99 | 32%        |
> | ACS-I      | 3.93  | 6.09 | 55%        |
> | ACS-E      | 3.53  | 4.25 | 20%        |
> | Average    | 2.91  | 3.97 | 36%        |
>
>
> The results show that, on average, YODO only results in a 36% increase in training time. When compared to an arbitrary accuracy-fairness trade-off, this extra time is negligible. For example, if we require 100 levels of trade-off on the ACS-E dataset, YODO takes 2.91 seconds, while training 100 fixed models takes 397 seconds. In this sense, the 2.91 needed by YODO is considered negligible.
>
>
>
>
>
> **4. Rewrite observation 9, as providing instance-level explanation is not a contribution of this work. The contribution is not having to train several models to achieve this, which was already discussed in previous paragraphs.** \
> We thank you for the insightful comments. We agree that providing the instance-level explanation is not a unique contribution of our work. Our method is more easy to provide instance-level explanation *with only one model*, while fixed training models need multiple models (which is typically infeasible in real-world applications).
>
> Thus, we rewrite Observation 9 to that *YODO can provide instance-level explanation for group fairness with only one model, while fixed training models need multiple models.* And we also revise the according explanation.
>
>
>
> **5. Please move the literature review earlier in the manuscript.** \
> Thank you for your valuable suggestion. We have moved the literature review to an earlier section, now in Section 2.

---

> ### Author Response · Authors · 2023-04-10
> **Response to Reviewer bMqY [2/3]**
>
> ### Concerns and Questions
>
> **1. Why do you train the "fair weights" on the regularized objective with $A=1$. It seems rather arbitrary, and it you want to be as fair as possible, I would at least increase  quite a bit.**
>
> We appreciate your insightful comment. We agree that the value of $A$ is arbitrary, and a larger value of $A$ is indeed expected to be fairer. However, larger values of $A$ usually also significantly decrease the accuracy of the downstream task. To investigate the effect of $A$, we conducted an experiment using YODO with varying values of $A$ ranging from [1, 5] and reported the results in Section 6.7 and Appendix C.2. The results show that: 1) as the value of $A$ increases, the fairness performance improves, but the accuracy of the downstream task deteriorates. 2) setting $A=1$ effectively addresses the trade-off between model accuracy and fairness, achieving an optimal balance.
>
>
>
> **2. Are the plots in Figure 2 and 3 are  correct? It seems rather odd that (to my understanding) there is no accuracy-fairness trade-off there at all: the most accurate models are always the fairest ones. Pareto fronts of this kind usually look like a concave curve, just like the ones of the MOO papers I referenced above.** \
>
> --*Update: We noticed that you edited your review. We thank you for your careful re-reading of our paper. We hope our response to your original review will address your concerns further.*--
>
> We thank you for the valuable suggestion. Figures 2 and 3 are correct, and they display the accuracy-fairness trade-off, as the optimal trade-off differs between our paper (**upper-right**) and the MOO paper (**lower-right**). In our case, if the optimal trade-off is in the upper-right corner, our Pareto fronts would appear as a concave curve. In our figures, the y-axis represents the error rate (lower is better), and the x-axis represents $\Delta DP$ (lower is better). Consequently, the upper-right corner signifies the best accuracy-fairness trade-off (while in Figure 1 of the MOO paper[1], the best trade-off is in the lower-right corner).
>
> [1] Scalable pareto front approximation for deep multi-objective learning. ICDM2021
>
>
> **3. Some aspects of the paper are unclear to me. For example, what hidden representation are we measuring exactly in Fig. 5.**
>
> We thank you for this insightful comment! We would like to clarify your concerns about the hidden representation (now Figure 6), and we have also revised Section 6.4 accordingly.
>
> Demographic parity requires that model predictions be independent of sensitive attributes (e.g., demographic groups). It is natural and rational to examine the independence between the hidden representation (penultimate layer) and the sensitive attribute. To interpret this representation, if $\alpha=0$ (unfair model), the representations of the male and female groups would contain more sensitive information (confirmed by the first figure on the bottom row in Figure 6). In contrast, if $\alpha=1$ (fair model), the representations would display a mix, signifying minimal sensitive information (confirmed by the last figure on the bottom row in Figure 6).
>
> Probing representations is also a common approach to examining fairness [1][2].
>
> [1] The variational fair autoencoder, ICLR2016 \
> [2] Fairness via Representation Neutralization, NeurIPS2021

---

> ### Author Response · Authors · 2023-04-10
> **Response to Reviewer bMqY [3/3]**
>
> **4. The paper makes underlying assumptions (e.g. that a trade-off between accuracy and fairness is needed), which would justify some design choices like the regularizer , but at the moment are not justified. For this particular example, why would I want the model weights to be orthogonal? What if the model could find the best accuracy while being fair?**
>
> We thank you for the thoughtful comment. We clarify your concern about the accuracy-fairness trade-off and the orthogonal weights.
>
> **Trade-off** As confirmed and substantiated by previous work[1][2][3][4], the trade-off between accuracy and fairness is a widely existing phenomenon for many fairness definitions. In our experiments, the results of fixed training (yellow lines in Figures 2 and 3) also demonstrate the trade-off between accuracy and fairness.
>
>
> **Orthogonal weights** Minimizing $\mathcal{L}\_{reg}$ (i.e., the cosine similarity of $\omega\_1$ and $\omega\_2$) promotes diversity (orthogonality) between the two endpoints. Given that the cosine similarity of two random high-dimensional vectors is typically close to zero, and considering that the accuracy-optimum and fairness-optimum points in the weight space are generally distinct, chasing $\mathcal{L}\_{reg} = \frac{ \langle \omega\_1, \omega\_2 \rangle^2 }{ | \omega\_1 |\_{2}^2 | \omega_2 |\_{2}^2 } = 0$ ensures that the two points will not be identical. We note that accuracy-optimum and fairness-optimum points becoming identical may not happen in experiments, we merely use this regularization term to prevent it from happening. We also revise Section 4.2 to further discuss the $\mathcal{L}\_{reg}$.
>
>
> [1] FACT: A diagnostic for group fairness trade-offs, ICML2020 \
> [2] Unlocking fairness: a trade-off revisited, NeurIPS2019 \
> [3] Is there a trade-off between fairness and accuracy? a perspective using mismatched hypothesis testing, ICML2020 \
> [4] Accuracy and fairness trade-offs in machine learning: A stochastic multi-objective approach, Computational Management Science 2022
>
>
> Thank you again for reviewing our paper. We hope our response and the revised paper address your concerns, and we are happy to answer any further questions you may have.
>
> Sincerely,\
> Authors

---

### Review · Reviewer_zQ7a · 2023-03-26

**Summary Of Contributions:**

This submission proposes to train two sets of neural network weights, one with cross-entropy loss and one with a combination of cross-entropy and a fairness loss, by computing gradients with respect to interpolations between them. An additional regularization term encourages low cosine similarity between the two sets of weights to be low. The method is evaluated on a handful of tabular datasets as well as CelebA, where it sometimes performs better than training with a fixed weighting for the fairness loss. The submission provides some additional experiments analyzing how the distribution of the model's output changes with $\alpha$.

**Audience:**

Yes

**Broader Impact Concerns:**

There is currently no broader impact statement, but since the work is intended to address fairness and the benefits of doing so are clearly described in the introduction, I do not think one is necessary.

**Claims And Evidence:**

Yes

**Requested Changes:**

- Because it's counterintuitive, I'd like to be more convinced that the proposed method actually provides better tradeoffs between accuracy and fairness than fixed training, even when all hyperparameters are tuned. Evidence could come in many forms. Comparing to numbers in previous work that provides well-tuned baselines could work. Performing a large hyperparameter sweep for at least one dataset and demonstrating that the findings do not change would also be sufficient. Providing some explanation of why this should happen might also be enough, if that explanation is convincing. I like the paper, but I feel that this needs to be addressed.
- The paper should provide full hyperparameters for training (learning rate, learning rate schedule if used, weight decay, etc.). This is critical.
- I'd like to see further analysis of the utility/necessity of $\mathcal{L}_{reg}$. This is non-critical; if the authors didn't do any experiments to verify its necessity then simply stating that would be fine.

**Strengths And Weaknesses:**

Strengths:

- The work is clearly motivated, and the method and experiments are described sufficiently clearly that it would be easy to reproduce them (modulo the missing hyperparameters).
- The proposed method is quite simple, yet seems effective.
- The proposed method seemingly sometimes Pareto-dominates training with a fixed weight for the fairness loss.

Weaknesses:

- Results are not directly compared with those achieved by previous work. The fact that this method sometimes outperforms a fixed training baseline is quite interesting. It would be more convincing if there are numbers in previous work that the authors could demonstrate they improve upon. Otherwise it's difficult to know whether further tuning would result in different patterns.
- The utility of $\mathcal{L}_{reg}$ for this task is not clear to me, and there are no empirical results to demonstrate its importance. In Wortsman et al. (2021), the regularization is needed to ensure that neural networks that lie within the subspace are diverse. Here, since networks with different values of $\alpha$ are not being ensembled, it is not obvious that diversity is important, and the loss itself should force the two points not to collapse.
- Full training hyperparameters (learning rate, weight decay, value of $\mathcal{L}_{reg}$) are not provided.
- In Figure 9, the minimum $\Delta$EO obtained by fixed training is sometimes much smaller than those obtained by the proposed method (e.g. for KDD Census and ACS-I-1Y Race), suggesting that there may be situations where, if one's goal is to achieve a high degree of fairness, fixed training is still necessary.
- Experiments are conducted only for small-ish experiments on simple networks (two-layer MLPs for small tabular datasets and a ResNet-18 on CelebA).

Minor:

- Forgive me if I have missed it, but $\\mathcal{L}_f$ does not appear to be explicitly introduced.
- Near the top of p. 6, the subscript for the 2-norm in the expression for $\\mathcal{L}_{reg}$ is not subscripted, and the "ce" and "f" in the final objective are also not subscripted.
- Axes in Figure 5 presumably have some meaning, but this meaning is not labeled.

Questions:

- Is there a difference between fixed training with $A=0$ and ERM? The submission says that the fixed training points are in $[0, 1]$ and there are 21 of them, which makes it sound like they should cover $A=0$, but the fixed training points do not appear to reach the ERM point.

---

> ### Author Response · Authors · 2023-04-10
> **Response to Reviewer zQ7a [1/2]**
>
> Dear Reviewer zQ7a,
>
> We thank you for your time and effort spent reviewing our paper. We are glad that you recognize the clear motivation, simple and effective method, and improved performance. We also appreciate your suggestions for our paper. We have updated our paper based on your comments and marked the changes in ${\color{orange} orange}$ color.
>
>
> ### Requested Changes
> **1. ...Comparing to numbers in previous work that provides well-tuned baselines could work. Performing a large hyperparameter sweep for at least one dataset and demonstrating that the findings do not change would also be sufficient. Providing some explanation of why this should happen might also be enough if that explanation is convincing. I like the paper, but I feel that this needs to be addressed.**
>
> We thank you very much for liking our paper, and we truly appreciate your valuable and constructive suggestions on how we can provide more convincing evidence. We make the following changes in our manuscripts:
> 1. We added more baseline methods.
> 2. The better performance of our proposed is attributed to the double trainable weights than fixed training. The doubled trainable weights would improve the expressive power of the neural network. Thus it would obtain a better accuracy-fairness trade-off than fixed training in most cases. Actually, in some cases (e.g., CelebA-Smilling-Gender shown in the 3rd subfigure in Figure 4).
> 3. We would like to clarify the results in Figure 3, we only use one model while fixed training needs $21$ model trained from scratch. This is the huge advantage of our method.
>
> **2. The paper should provide full hyperparameters for training (learning rate, learning rate schedule if used, weight decay, etc.). This is critical.** \
> Thank you for your valuable suggestion. We added more hyperparameters for our experiments in Appendix B.4 on Page 23 in the revised paper.
>
>
> **3. I'd like to see further analysis of the utility/necessity of $\mathcal{L}\_{reg}$. This is non-critical; if the authors didn't do any experiments to verify its necessity then simply stating that would be fine.** \
> We thank you for this insightful suggestion. Minimizing $\mathcal{L}\_{reg}$ (i.e., the cosine similarity of $\omega\_1$ and $\omega\_2$) promotes diversity between the two endpoints. Given that the cosine similarity of two random high-dimensional vectors is typically close to zero, and considering that the accuracy-optimum and fairness-optimum points in the weight space are generally distinct, chasing $\mathcal{L}\_{reg} = \frac{ \langle \omega\_1, \omega\_2 \rangle^2 }{ | \omega\_1 |\_{2}^2 | \omega_2 |\_{2}^2 } = 0$ ensures that the two points will not be identical. We note that accuracy-optimum and fairness-optimum points becoming identical may not happen in experiments, we merely use this regularization term to prevent it from happening.
>
> We have also revised Section 4.2 on Page 7 to provide a more in-depth discussion on the functionality of $\mathcal{L}\_{reg}$.
>
>
>
>
> ### Concerns and Questions
>
> **1. In Figure 9, the minimum EO obtained by fixed training is sometimes much smaller than those obtained by the proposed method (e.g. for KDD Census and ACS-I-1Y Race), suggesting that there may be situations where, if one's goal is to achieve a high degree of fairness, fixed training is still necessary.** \
> We thank you for these insightful comments. We agree that the proposed method may not always achieve the minimum EO score that can be obtained through fixed training. In some other cases, our method has a large range of fairness metrics, such as in most cases in Figure 3, on Demographic Parity.
>
>
>
> **2. Experiments are conducted only for small-ish experiments on simple networks (two-layer MLPs for small tabular datasets and a ResNet-18 on CelebA).** \
> We thank you for your valuable suggestions. To clarify your concerns, we conducted experiments on larger models on the CelebA dataset, specifically examining ResNet18, ResNet34, ResNet50, ResNet101, WideResNet50, and WideResNet101. The results show that 1) all models were capable of achieving a flexible accuracy-fairness trade-off during inference.
> 2) Larger models demonstrated greater stability. The results show that our models works can also work on larger models.
> We have added these experiments to Appendix C.4 on Page 26.

---

> ### Author Response · Authors · 2023-04-10
> **Response to Reviewer zQ7a [2/2]**
>
> **3. Forgive me if I have missed it, but $\mathcal{L}\_{f}$ does not appear to be explicitly introduced.** \
> We thank you for bringing this to us. We unintentionally did not explain $\mathcal{L}\_{f}$ when it first appeared in Section 3.1. In our original paper, we introduced it in Section 4.1 (below Equation (3), second occurrence) as *"$\mathcal{L}\_{f}$ is instantiated as the demographic parity difference $\Delta \text{DP}[f(x; \theta)]$*."  We have added an introduction to $\mathcal{L}\_{f}$ in Section 3.1 in our revised paper.
>
>
> **4. Near the top of p. 6, the subscript for the 2-norm in the expression for  is not subscripted, and the "ce" and "f" in the final objective are also not subscripted.** \
> Thanks for your careful reading. We have carefully reviewed the text and corrected these formatting errors in our revised paper. Thank you for helping us improve the clarity of our work. The revised paragraph is on Page 7 in the updated paper.
>
>
>
>
> **5. Axes in Figure 5 presumably have some meaning, but this meaning is not labeled.** \
> Thank you for your valuable suggestions. We have updated Figure 6 (previously Figure 5 in the original paper) and Figure 13 to include labels for the X-Axis and Y-Axis. The X-Axis represents the predictive values $\hat{y}$ and the Y-Axis represents the density (normalized frequency) for different predictive values.
>
>
>
> **6. Is there a difference between fixed training with $A=0$ and ERM? The submission says that the fixed training points are in $[0,1]$ and there are 21 of them, which makes it sound like they should cover $A=0$, but the fixed training points do not appear to reach the ERM point.** \
> Thank you for your careful reading and for bringing this to us. The fixed training with $A=0$ and ERM are indeed the same. In fact, the reported result of ERM was conducted using fixed training with $A=0$.
>
> To ensure a more accurate presentation, we revise the description of fixed training in Section 6 on Page 8 *with different values of $A$, which is set to $[0,1]$ with an interval of $0.05$* to *with different values of $A$, which is set to $(0,1]$ with an interval of $0.05$*.
>
>
>
> Thank you again for reviewing our paper. We hope our response and the revised paper address your concerns, and we are happy to answer any further questions you may have.
>
> Sincerely,\
> Authors

---

### Review · Reviewer_DMiX · 2023-03-27

**Summary Of Contributions:**

The authors present a method for fairness in machine learning that allow flexibility in adjusting the tradeoff between accuracy and fairness. Unlike many previous fair ML models that require the tradeoff to be set during the training time, the proposed mode has the capability to set the tradeoff at inference time without requiring to re-train the model. The proposed method is based on the neural network subspace models (Wortsman et al. (2021) and Benton et al. (2021)) where it tries to find an objective-diverse subspace (line), where on one side fully optimizes for accuracy, where the other side focuses on making a fair prediction. The tradeoff between accuracy and fairness can then be set at inference time by combining both sides of the "line" using a balancing variable $\alpha$.
The authors derive a training procedure for the model. Finally, the authors demonstrate the effectiveness of the models in several tabular and image datasets.

**Audience:**

Yes

**Broader Impact Concerns:**

I do not have any broader impact concerns for this paper.

**Claims And Evidence:**

Yes

**Requested Changes:**

- Please add a section explaining the prediction procedure
- Add additional baselines. Particularly some recent fair models.
- Discussion and experiments on other fairness criteria.
- Fix some presentation issues.


**Strengths And Weaknesses:**

Strengths:
- The flexibility in choosing accuracy-fairness tradeoffs at inference time benefits many real-world prediction scenarios.
- The use of neural network subspace for accuracy-fairness tradeoffs in the proposed model is interesting.
- The experiments are conducted in both tabular and image datasets.
- The results show that the proposed method is competitive over the baseline that sets the tradeoff during training.
- The authors provide detailed analyses of the experiment results


Weaknesses:
- Contributions. The proposed method is a rather direct application of the neural network subspace for modeling accuracy-fairness tradeoffs.
- Model. The authors describe in detail about the training procedure. But the paper lacks any description of the prediction procedure. Given the learned weights, how does the prediction needs to be done for a certain value of $\alpha$?
- Baselines. The baselines the authors use are limited. Only a similar network architecture with a fixed accuracy-fairness tradeoff, set during the training procedures. There are so many fairness models available out there. It will be useful if the authors include some recent fair models as the baselines in the experiments.
- Fairness criteria. The authors only focus on models and experiments for demographic parity criteria. I suggest the authors also add discussion and conduct experiments on other fairness criteria, such as equalized odd and equalized opportunity.
- Presentation. The presentation of the figures in Figure 2 and Figure 3 is a bit unconventional. The x-axis runs from to bottom rather than the usual bottom up. This makes it a bit harder to interpret the results.
- Presentation [Minor]: Undefined citation in Section 2.1,  the paragraph after Definition 2.1.

---

> ### Author Response · Authors · 2023-04-10
> **Response to Reviewer DMiX**
>
> Dear Reviewer DMiX,
>
> Thank you for your time and effort in reviewing our paper. We appreciate your recognition of the flexibility in accuracy-fairness tradeoffs, the interesting use of neural network subspace, and the thorough experiments conducted on various datasets. We are also grateful for your acknowledgment of our method's competitiveness and detailed analyses. Your support is highly encouraging. In the following, we address your concerns about the weaknesses and the requested changes and we also revised the paper labeled in ${\color{purple} purple}$ color.
>
>
>
>
> ### Requested Changes and Concerns
> **1. Please add a section explaining the prediction procedure** \
> Thank you for your great suggestion! We have added Section 4.3 on Page 7 to our revised paper, which explains the prediction procedure in detail. Additionally, we have included an illustrative figure (Figure 2 on Page 7) to further clarify the prediction procedure. The brief prediction procedure is as follows:
>
> 1. Choose the desired trade-off parameter $\alpha$, which controls the balance between accuracy and fairness.
> 2. Compute the weighted combination of the two sets of trained weights, $(1-\alpha)\omega_1 + \alpha\omega_2$, to obtain the model parameters for the desired trade-off.
> 3. Compute the prediction function to the test sample $\mathbf{x}$ as $f(\mathbf{x}; (1-\alpha)\omega_1 + \alpha\omega_2)$, to obtain the predicted output.
>
> Note that the prediction procedure enables users to choose the desired level of accuracy and fairness for their specific application by choosing desired trade-off parameter $\alpha$.
>
>
>
> **2. Add additional baselines.** \
> Thank you for your valuable suggestions. To further confirm the efficacy of the proposed method, we incorporated additional widely-adopted baselines for demographic parity and presented the new results in Figure 3 on Page 9, such as: 1) Prejudice Remover[1] 2) Adversarial Debiasing[2][3]. The results still demonstrate the effectiveness of the proposed method.
>
> [1] Fairness-aware classifier with prejudice remover regularizer, ECML-PKDD 2012 \
> [2] Learning to pivot with adversarial networks, NeurIPS2017 \
> [3] Learning adversarially fair and transferable representations, ICML2018
>
>
>
> **3. Discussion and experiments on other fairness criteria.** \
> Thank you for your valuable suggestions. In Section 6.6 (previously Section 5.6), we conducted experiments on another group fairness measure, Equality of Opportunity. Based on your recommendation, we included experiments on Equalized Odds and presented the experimental results in Appendix C.2 on Page 24. Through experiments on multiple fairness definitions (i.e., DP, EO, EOdd), we can gain a more comprehensive understanding of our approach's performance and the trade-offs between fairness and accuracy.
>
>
>
> **4. Fix some presentation issues.** \
> We thank you for this great suggestion!
>
> We agree that Figures 2, 3, 4 and 9 might be unconventional, which is the result that we try our best to make the results easier to understand. We report the Error Rate and $\Delta DP$, where lower values indicate better performance. In doing so, the *upper-left* would be the best accuracy-fairness trade-off.
>
> We appreciate your careful reading of our paper. For the undefined citation in Section 3.1 (previously Section 2.1), we found that we mistakenly cited a paper [1] twice and have fixed it.
>
>
> [1] Optimized score transformation for fair classification, ICML2020
>
>
>
> We thank you for reviewing our paper once again. We hope our response and the updated paper address your concerns, and we are happy to answer any further questions you may have.
>
> Sincerely,\
> Authors

---

### Review · Reviewer_21XQ · 2023-03-28

**Summary Of Contributions:**

This paper presents a new method to smoothly trade off fairness and accuracy. In particular, rather than training a single network with a weighted accuracy-fairness loss, the paper proposes to learn two different networks: one that is accurate (but potentially unfair) and another that is fair (but potentially inaccuate). The paper then leverages a recent line of work on weight interpolation, showing that linearly mixing the weights allows one to achieve a variety of different accuracy-fairness tradeoffs at test time.

**Audience:**

Yes

**Broader Impact Concerns:**

I have no broader impact concerns, but as previously mentioned I do think that the paper would benefit from either scoping down its claims about "fairness" or substantiating them beyond DP and one EO experiment.

**Claims And Evidence:**

Yes

**Requested Changes:**

- Additional experiments on other notions of fairness or better scoping of the claims / removal of "without loss of generality" argument
- Fully described experimental details so that the work can be reproduced
- Baselines: I am not an expert in this space, but it seems like this paper cites many other papers that also try to find a pareto-optimal combination of accuracy and fairness. The authors should either compare against these methods or explain why they are inapplicable.
- Relatedly, a more thorough discussion of related works, especially those that explicitly aim to find the accuracy-fairness pareto frontier.
- Removing/further substantiating imprecise claims


**Strengths And Weaknesses:**

Strengths:
- The motivation for the paper is quite clear---it is often hard in practice to commit to a fixed fairness/accuracy tradeoff, and so being able to "tune" this accuracy at test time is quite valuable.
- The method is simple and easy to implement, and also builds on a line of related works that suggest weight averaging as a promising avenue for combining machine learning models.
- The results are promising: YODO is never dominated by training with a fixed accuracy-fairness coefficient, despite the latter being much less flexible at test time. In fact, there are some datasets where YODO seems to be strictly better than training with a fixed coefficient.

Weaknesses:
- The paper examines a relatively narrow definition of "fairness" (demographic parity), which is fine but the paper should either experiment with other notions of fairness or appropriately scope the claims made. In particular, I'm not sure I see why this is "without loss of generality," given that the main results are experimental---in fact, looking at the preliminary experiments on EO, they seem qualitatively very different from the ones on DP.
- Experiments:
  - There are no experimental details with which one can replicate the results.
  - The paper compares with (and impressively beats) a fixed fairness-accuracy coefficient. What about other methods for ensuring demographic parity?
- The writing is quite confusing in many places, including many of the section titles containing grammatical errors. This did not affect my rating of the paper, but I think that proofreading further would improve clarity.
- I'm not sure I understood the jump between (3) and (4) - why do we need to sample alpha randomly? Why can't one train one model with alpha = 0 and another model with large alpha and interpolate between these two models?
- Relatedly, the choice of alpha = 1 to be the "fair classifier" seems somewhat arbitrary, given that they are both just loss functions, couldn't alpha >> 1 be "more fair"? When comparing to a fixed accuracy-fairness tradeoff, did the authors try using alpha > 1?
- The paper is also prone to making some imprecise claims (or at least, claims that should be further explained). The "without loss of generality" claim is one, the argument going from (3) to (4) is another, but there are many more (e.g., "Our model can be regarded as training infinite models with different fairness constraints" - it is unclear to me what this means).

---

> ### Author Response · Authors · 2023-04-10
> **Response to Reviewer 21XQ [1/2]**
>
> Dear Reviewer 21XQ,
>
> We sincerely appreciate your time and effort in reviewing our paper. We are glad that you recognized the clear motivation, simple method, and promising results of our work. We are also grateful for your constructive suggestions on the presentation and experiments in our paper and highlighted in ${\color{green} green}$ color.
>
>
>
> ### Requested Changes
> **1. Additional experiments on other notions of fairness or better scoping of the claims / removal of "without loss of generality" argument** \
> We thank you for this valuable comments. We remove the argument that"without loss of generality". In addition to the used fairness notions (Demographic Parity and Equality of Opportunity) in our submission, we added the experiments on Equalized Odds in the updated version. The results are presented in Section 6.6.
>
>
>
> **2. Fully described experimental details so that the work can be reproduced** \
> We thank you for this valuable comment. We have revised Appendix B.4 on Page 23 to add more detailed experimental settings.
>
>
>
> **3. Baselines:it seems like this paper cites many other papers that also try to find a pareto-optimal combination of accuracy and fairness. The authors should either compare against these methods or explain why they are inapplicable.** \
> Thank you for your valuable suggestions. To further confirm the efficacy of the proposed method, we incorporated additional widely-adopted baselines for demographic parity and presented the new results in Figure 3 on Page 9, such as: 1) Prejudice Remover[1] 2) Adversarial Debiasing[2][3]. The results still demonstrate the effectiveness of the proposed method.
>
> [1] Fairness-aware classifier with prejudice remover regularizer, ECML-PKDD 2012 \
> [2] Learning to pivot with adversarial networks, NeurIPS2017 \
> [3] Learning adversarially fair and transferable representations, ICML2018
>
>
>
> **4. Relatedly, a more thorough discussion of related works, especially those that explicitly aim to find the accuracy-fairness pareto frontier.** \
> We thank you for your valuable suggestion. We have added more discussions of related works in Section 2. We included discussions of papers related to the accuracy-fairness trade-off, such as [1], [2], [3], and [4]. Additionally, we have expanded the discussion on neural network spaces.
>
>
> [1] FACT: A diagnostic for group fairness trade-offs, ICML2020 \
> [2] Unlocking fairness: a trade-off revisited, NeurIPS2019 \
> [3] Is there a trade-off between fairness and accuracy? a perspective using mismatched hypothesis testing, ICML2020 \
> [4] Accuracy and fairness trade-offs in machine learning: A stochastic multi-objective approach, Computational Management Science 2022
>
>
>
>
> **5. Removing/further substantiating imprecise claims** \
> We thank you for the comment to improve our work. We make the following changes to clarify your concerns:
>
> 1. We remove the argument "without loss of generality" and add more baseline methods.
> 2. We clarify the need for the argument going from (3) to (4) in Concern 2. And we also add discussion and experiment for this in Appendix C.1 on Page 24..
> 3. We add a further explanation for the argument: *Our model can be regarded as training infinite models with different fairness constraints.*, as provided in Concern 3. We also revise the paper in Section 4.1.
>
>
> ### Concerns and Questions
> **1. The writing is quite confusing in many places, including many of the section titles containing grammatical errors. This did not affect my rating of the paper..**
>
> Thank you for your feedback regarding the writing and grammatical errors in the paper. We have thoroughly revised the manuscript and ensured that all grammatical errors are corrected in our updated version.

---

> ### Author Response · Authors · 2023-04-10
> **Response to Reviewer 21XQ [2/2]**
>
>
> **2. ...between (3) and (4) - why do we need to sample alpha randomly? Why can't one train one model with alpha = 0 and another model with large alpha and interpolate between these two models?** \
> **3. e.g., "Our model can be regarded as training infinite models with different fairness constraints" - it is unclear to me** \
> We thank you for these two great comments! Since they (2 and 3) are essentially the same question, we will answer them together below.
>
> Training two models separately and interpolating them can not achieve our goal - flexible accuracy-fairness trade-offs at inference time. As verified and studied by previous works [1][2][3][4], interpolating the weights $\omega_1$ and $\omega_2$ of two separate well-trained models at inference time has been shown to achieve no better accuracy than an untrained model. We also added an experiment to verify this point in our case and we present the results in Appendix C.1. The results show that interpolating the weights $\omega_1$ and $\omega_2$ of the two separate well-trained models at inference time achieves worse accuracy, since the error rate is relatively high when $\alpha=0.5$.
>
>
> Since our method aims to find the "line" between the accuracy-optimum and fairness-optimum points in the weight space, we need to ensure that any point (infinite models) on this line corresponds to a specific level of fairness. Therefore, we randomly sample an $\alpha$ during each epoch. In other words, each $\alpha$ corresponds to one model, and with a wide range of different $\alpha$ values, we train numerous models (infinite) throughout the training process.
>
> [1] Linear mode connectivity and the lottery ticket hypothesis. ICML2020 \
> [2] Learning neural network subspaces. ICML2021 \
> [3] Deep learning versus kernel learning: an empirical study of loss landscape geometry and the time evolution of the neural tangent kernel. NeurIPS2020 \
> [4] Loss surface simplexes for mode connecting volumes and fast ensembling. ICML2021.
>
>
>
> **4. the choice of alpha = 1 to be the "fair classifier" seems somewhat arbitrary, given that they are both just loss functions, couldn't alpha >> 1 be "more fair"? When comparing to a fixed accuracy-fairness tradeoff, did the authors try using alpha > 1?**
>
> ----*We think the alpha you mentioned is $A$ in Equation 2, which is set to $1$. Please correct us if we are wrong*---
>
> We appreciate your insightful comment. We agree that the value of $A$ is arbitrary and a larger value of $A$ is indeed expected to be fairer. However, larger values of $A$ usually also significantly decrease the accuracy of the downstream task. To investigate the effect of $A$, we conducted an experiment using YODO with varying values of $A$ ranging from [1, 5] and reported the results in Section 6.7 and Appendix C.2. The results show that: 1) as the value of $A$ increases, the fairness performance improves, but the accuracy of the downstream task significantly deteriorates. 2) setting $A=1$ effectively addresses the trade-off between model accuracy and fairness, achieving an optimal balance.
>
>
>
> We thank you for reviewing our paper again. We hope our response and the updated version address your concerns. We are happy to answer any further questions you may have.
>
> Sincerely,\
> Authors

---

### Author Response · Authors · 2023-04-10
**General Response**

Dear Reviewers,

We sincerely thank you all for reviewing our paper and providing insightful feedback. Your comments have helped us significantly improve our work. We are glad that our proposed idea, motivation, and writing quality have been well-received. We acknowledge concerns about typos, assumptions, literature review, and comparisons with existing methods, as well as the need to clarify specific aspects of the paper.

We have addressed these concerns and refined our submission based on your feedback. The revisions addressing the concerns of Reviewer ${\color{purple} DMiX}$, ${\color{orange} zQ7a}$,${\color{blue} bMqY}$ and ${\color{green} 21XQ}$  have been highlighted in different colors in our updated paper. The main revisions are as follows:

1. For the experiments:
    - We added more baseline methods.
    - We conducted experiments on equalized odds, in addition to demographic parity and equality of opportunity.
    - We added experiments on different $A$ values.
    - We added experiments on time complexity.
    - We added experiments that trained the two endpoints separately.

2. For the concerns and questions:
    - We added more discussion about related papers, including accuracy-fairness trade-off and neural network subspace.
    - We provided more details about the experimental setting.
    - We revised the discussion of our method in Section 4.2 to fix the backward order of $\omega_1$ and $\omega_2$.
    - We added the prediction procedure of YODO as well as a figure to illustrate the procedure.
    - We fixed all the typos that reviewers pointed out and we found.


We hope our response address your concerns, and we thank you for your valuable comments.

Once again, we thank all the reviewers for their time and effort in reviewing our paper. And we are happy to answer any further questions you may have.

Sincerely, \
Authors

---

### Decision · Action_Editors · 2023-05-08

**Recommendation:** Reject

**Comment:**

This paper presents a new method to smoothly trade off fairness and accuracy. In particular, rather than training a single network with a weighted accuracy-fairness loss, the paper proposes to learn two different networks: one that is accurate (but potentially unfair) and another that is fair (but potentially inaccuate). The paper then leverages a recent line of work on weight interpolation, showing that linearly mixing the weights allows one to achieve a variety of different accuracy-fairness tradeoffs at test time.

We appreciate the efforts from authors in their detailed rebuttal. Given the clear motivation, the simplicity of the proposed approach and the strong empirical evidence, I personally love this idea. However, two qualified reviewers still tend to reject this work after reading rebuttal, while one qualified reviewer keeps positive. Therefore, the current version still has some problems. For example, (1) Even though the authors have provided additional baselines, only the tabular datasets experiments have them. The other experiments do not have additional baselines. (2) The additional baselines provided are from older papers (2012, 2017, and 2018). It will be good to use more recent baselines. (3) The authors also add experiments for Equalized Odd metric, but the experiments only use Fixed Training baselines. More baselines are suggested as well. (4) This method is a specific case of the solution proposed in "You Only Train Once". The similarities in the titles could've been a coincidence, but after pointing out to the authors, they just limited to put them in the related work as another work, without really pointing out the similarities and giving them credit.

Therefore, we cannot accept this work this time, but the authors are highly encouraged to **resubmit after a major and significant revision**. We will consider to recommend its acceptance if the authors had addressed these issues properly.

**Audience:**

Yes

**Claims And Evidence:**

The current version still has some problems. For example, (1) Even though the authors have provided additional baselines, only the tabular datasets experiments have them. The other experiments do not have additional baselines. (2) The additional baselines provided are from older papers (2012, 2017, and 2018). It will be good to use more recent baselines. (3) The authors also add experiments for Equalized Odd metric, but the experiments only use Fixed Training baselines. More baselines are suggested as well. (4) This method is a specific case of the solution proposed in "You Only Train Once". The similarities in the titles could've been a coincidence, but after pointing out to the authors, they just limited to put them in the related work as another work, without really pointing out the similarities and giving them credit.